# A broadly neutralizing humanized ACE2-targeting antibody against SARS-CoV-2 variants

Yanyun Du[1,10], Rui Shi[2,10], Ying Zhang[3,10], Xiaomin Duan[2,4], Li Li[5], Jing Zhang[5], Fengze Wang [2,4], Ruixue Zhang[3], Hao Shen[3], Yue Wang [2,4], Zheng Wu[2,6], Qianwen Peng[1], Ting Pan[1], Wanwei Sun[1], Weijin Huang [7], Yue Feng [8], Hui Feng[5], Junyu Xiao [3], Wenjie Tan [9,11✉], Youchun Wang [7,11✉], Chenhui Wang [1,11✉] & Jinghua Yan [2,4,11✉]

The successive emergences and accelerating spread of novel severe acute respiratory syndrome coronavirus 2 (SARS-CoV-2) lineages and evolved resistance to some ongoing clinical therapeutics increase the risks associated with the coronavirus disease 2019 (COVID-19) pandemic. An urgent intervention for broadly effective therapies to limit the morbidity and mortality of COVID-19 and future transmission events from SARS-related coronaviruses (SARSr-CoVs) is needed. Here, we isolate and humanize an angiotensin-converting enzyme-2 (ACE2)-blocking monoclonal antibody (MAb), named h11B11, which exhibits potent inhibitory activity against SARS-CoV and circulating global SARS-CoV-2 lineages. When administered therapeutically or prophylactically in the hACE2 mouse model, h11B11 alleviates and prevents SARS-CoV-2 replication and virus-induced pathological syndromes. No significant changes in blood pressure and hematology chemistry toxicology were observed after injections of multiple high dosages of h11B11 in cynomolgus monkeys. Analysis of the structures of the h11B11/ACE2 and receptor-binding domain (RBD)/ACE2 complexes shows hindrance and epitope competition of the MAb and RBD for the receptor. Together, these results suggest h11B11 as a potential therapeutic countermeasure against SARS-CoV, SARS-CoV-2, and escape variants.

[1] Key Laboratory of Molecular Biophysics of the Ministry of Education, National Engineering Research Center for Nanomedicine, College of Life Science and Technology, Huazhong University of Science and Technology, Wuhan, China. [2] CAS Key Laboratory of Microbial Physiological and Metabolic Engineering, Institute of Microbiology, Chinese Academy of Sciences, Beijing, China. [3] State Key Laboratory of Protein and Plant Gene Research, School of Life Sciences, Peking-Tsinghua Center for Life Sciences, Beijing Advanced Innovation Center for Genomics, Peking University, Beijing, China. [4] University of Chinese Academy of Sciences, Beijing, China. [5] Shanghai Junshi Biosciences Co., Ltd, Shanghai, China. [6] Institute of Physical Science and Information, Anhui University, Hefei, China. [7] Division of HIV/AIDS and Sex-transmitted Virus Vaccines, Institute for Biological Product Control, National Institutes for Food and Drug Control (NIFDC) and WHO Collaborating Center for Standardization and Evaluation of Biologicals, Beijing, China. [8] Beijing Advanced Innovation Center for Soft Matter Science and Engineering, Beijing Key Laboratory of Bioprocess, State Key Laboratory of Chemical Resource Engineering, College of Life Science and Technology, Beijing University of Chemical Technology, Beijing, China. [9] NHC Key Laboratory of Biosafety, National Institute for Viral Disease Control and Prevention, Chinese Center for Disease Control and Prevention, Beijing, China. [10] These authors contributed equally: Yanyun Du, Rui Shi, Ying Zhang. [11] These authors jointly supervised this work: Wenjie Tan, Youchun Wang, Chenhui Wang and Jinghua Yan. ✉email: tanwj@ivdc.chinacdc.cn; wangyc@nifdc.org.cn; wangchenhui@hust.edu.cn; yanjh@im.ac.cn

Globally, in the past year, coronavirus disease 2019 (COVID-19) has emerged as an unprecedented public health emergency, with over 100 million confirmed cases and 2 million deaths (World Health Organization). Severe acute respiratory syndrome coronavirus 2 (SARS-CoV-2), the causative agent of the pandemic, is a positive-sense, single-stranded, enveloped RNA virus that belongs to the lineage B betacoronavirus family[1]. To mediate receptor recognition and membrane fusion, the spike (S) glycoprotein of SARS-CoV-2 binds to angiotensin-converting enzyme-2 (ACE2) on host cells, which has previously been proven to be the membrane-anchored receptor for both human respiratory coronavirus NL63 (HCoV-NL63) and SARSr-CoV (SARS-related coronavirus)[2,3].

Neutralizing antibodies (NAbs) are a promising class of therapeutics against SARS-CoV-2 infection, and Emergency Use Authorizations (EUAs) have been issued for four NAbs to treat COVID-19[4–6]. As the SARS-CoV-2 S protein depends upon the engagement of ACE2 by the receptor-binding domain (RBD) for cell entry, multiple studies have described the isolation and characterization of potent NAbs targeting the SARS-CoV-2-RBD[7]. However, inevitably, SARS-CoV-2-RBD has been shown to rapidly accumulate escape mutations under the strong selective pressure applied in the setting of therapeutics[8,9]. Such a resistance mechanism, which limits permanent responses to therapy and clinical applications, shows similarity to the viral escape through mutation of the HIV-1 envelope glycoprotein[10] and influenza hemagglutinin protein[11]. Importantly, CoVs have long been predicted to have a high probability of cross-species transmission and cause zoonotic disease and pandemics in humans[12,13]. Although SARS-CoVs and SARSr-CoVs belong to the same genera, share a high sequence identity among their S proteins, and recognize identical cellular receptors for virus entry, SARSr-CoV NAbs fail to potently cross-protect against infection with SARS-CoV-2 and SARS-CoV[2]. On the basis of this evidence, broadly effective therapies are needed as an urgent countermeasure to limit morbidity and mortality of COVID-19, the emergence of escape mutants of SARS-CoV-2, SARS-CoV re-emergence, and future zoonotic transmission events from SARSr-CoVs currently circulating in bat populations.

Monoclonal antibodies (MAbs) targeting viral receptors have been proved to block viral infections. Ibalizumab, an approved MAb that binds human CD4 to block HIV-1 infection[14], showed antiviral and immunologic activity in a phase 3 study[15]. Meanwhile, the results from Hoffman et al. showed that mouse-derived polyclonal antibodies targeting ACE2-blocked SARS-CoV-2 pseudovirus-infected host cells[16]. However, monoclonal antibody drugs targeting ACE2 have not been reported. In this study, we identify a broad-spectrum humanized ACE2/RBD-blocking MAb, h11B11, which demonstrates potent inhibitory activity against SARS-CoV and circulating global SARS-CoV-2 lineages in vitro. h11B11 inhibits SARS-CoV-2 infection in mouse models in both prophylactic and treatment settings. Importantly, no significant changes in blood pressure and hematology chemistry toxicology were observed after injections of multiple high dosages of h11B11 in cynomolgus monkeys. Furthermore, structural comparisons of h11B11/ACE2 with RBDs/ACE2 reveal the mechanisms of its antiviral activities. These assessments highlight h11B11 as a broad antiviral intervention aimed at CoVs.

## Results

**Isolation and identification of an ACE2-blocking MAb.** To generate murine anti-hACE2 antibodies, BALB/c mice received hACE2 (19–615) soluble antigens in a prime-boost immunization regimen with a 4-week interval. Using hybridoma technology, we obtained a number of mouse anti-hACE2 cell clones. After

screening hybridoma supernatants, five clones of the MAbs that blocked HEK293T-hACE2 cell infection with SARS-CoV and SARS-CoV-2 spike pseudotyped virus were identified (Supplementary Fig. 1a, b). Notably, clone h11B11 exhibited significant inhibitory activity against pseudotyped virus infection. The sequences of the variable regions of h11B11 in the hybridoma cell line were obtained through rapid amplification of complementary DNA (cDNA) ends amplification. To minimize immunogenicity, antibody-dependent cellular phagocytosis, and antibody-dependent cell cytotoxicity, murine h11B11 was humanized through complementarity determining region (CDR) grafting onto human acceptor germline frameworks and engineered as a human IgG$_4$ subclass. Surface plasmon resonance (SPR) showed that the binding kinetics of humanized and parental MAbs to hACE2 was consistent (Supplementary Fig. 1c and Fig. 1b). Interrupting the interaction between the virus and cellular receptors is an efficient approach to maximize the inhibition of causative agent entry into host cells. Meanwhile, ACE2 has been identified as the receptor for SARS-CoV and SARS-CoV-2, and these CoVs bind the overlapping region of ACE2, so we sought to validate whether h11B11 is an ACE2/RBD-blocking MAb that may be able to prevent SARSr-CoV-RBD binding to ACE2. The observation that up to 100% inhibition was achieved upon membrane-anchored hACE2 saturated with SARS-CoV-RBD or SARS-CoV-2-RBD proteins in the presence of h11B11 indicated that the MAb against hACE2 was targeted to the RBD-binding site (Fig. 1a). Furthermore, SPR showed that h11B11 has a higher affinity ($K_D = 2.95$ nM) than the SARS-CoV-2-RBD or SARS-CoV-2 S protein for immobilized hACE2 ($K_D = 36.4–133.3$ nM for SARS-CoV-2-RBD and ~15 nM for SARS-CoV-2 S)[2,17–19], almost entirely due to a slower off-rate (Fig. 1b). hACE2 polymorphisms were likely associated with the genetic susceptibility of COVID-19, and the distributions of 63 variants in different populations were systematically investigated[20]. The binding profiles of h11B11 to five hACE2 variants (S19P, I21T, K26R, N33D, and D38E) in the N-terminal helix (NTH), which may greatly affect the stability of receptors and MAb binding, were further analyzed. No substantial differences were observed in the binding affinities of h11B11 to wild-type hACE2 ($K_D = 2.95$ nM) and the four variants ($K_D = 3.49$ nM for I21T, $K_D = 1.58$ nM for K26R, $K_D = 1.63$ nM for N33D, and $K_D = 2.85$ nM for D38E) (Fig. 1c). However, the S19P variant exhibited complete resistance to the MAb, revealing the importance of S19 on NTH for h11B11 binding to hACE2. Therefore, the broad binding activity of h11B11 to the viral receptor is not affected by most nonsynonymous hACE2-NTH variants.

**Antibody h11B11 does not affect the carboxypeptidase activity of hACE2 in vitro.** ACE2 performs a multiplicity of physiological roles, including acting as a counterregulator of the renin–angiotensin system (RAS), a transporter of amino acids, and a receptor of CoVs[21]. The entry of SARSr-CoVs into host cells is facilitated by ACE2 receptors, followed by the downregulation of surface ACE2 expression[21]. It was reported that decreased ACE2 levels induced RAS imbalance and multiorgan damage, which suggests that balanced catalytic activity of ACE2 may also protect the lung from virus injury[22]. The physiological role of ACE2 is to act as a carboxypeptidase that potently degrades angiotensin II to angiotensin-(1–7)[21]. Therefore, we intended to explore whether the hACE2-blocking antibody would affect the carboxypeptidase activity of ACE2. To eliminate the nonspecific effect of protein on enzyme activity, hACE2 proteins treated with isotype IgG were employed as control. By using an in vitro hACE2 enzyme activity assessment, we found that the Michaelis–Menten constant ($K_m$) of the catalytic activity of

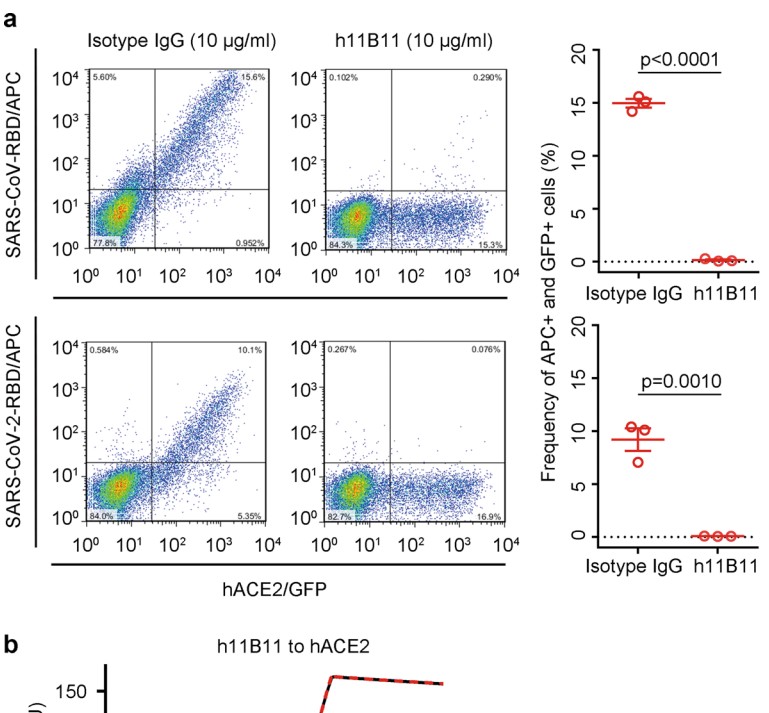

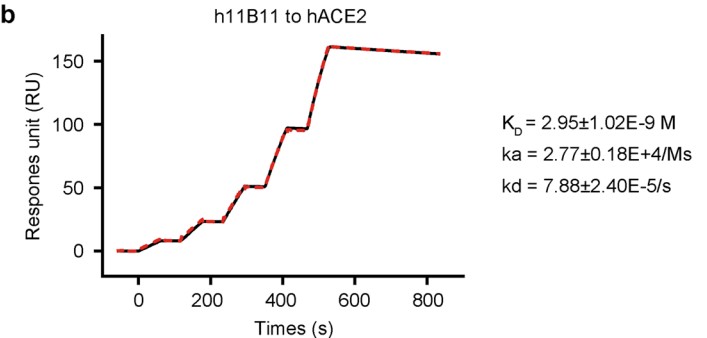

h11B11 to hACE2

$K_D = 2.95\pm1.02E\text{-}9$ M
ka = $2.77\pm0.18E\text{+}4$/Ms
kd = $7.88\pm2.40E\text{-}5$/s

**c**

|  | ka (1/Ms) | kd (1/s) | $K_D$ (M) |
|---|---|---|---|
| hACE2  wt | 2.77±0.18E+4 | 7.88±2.40E-5 | 2.95±1.02E-9 |
| hACE2 S19P | n.d | n.d | n.d |
| hACE2  I21T | 3.20±0.57E+4 | 10.71±0.69E-5 | 3.49±0.40E-9 |
| hACE2 K26R | 3.13±0.23E+4 | 4.92±1.95E-5 | 1.58±0.60E-9 |
| hACE2 N33D | 3.33±0.62E+4 | 5.36±1.20E-5 | 1.63±0.33E-9 |
| hACE2 D38E | 3.14±0.82E+4 | 4.91±2.45E-5 | 2.85±0.77E-9 |

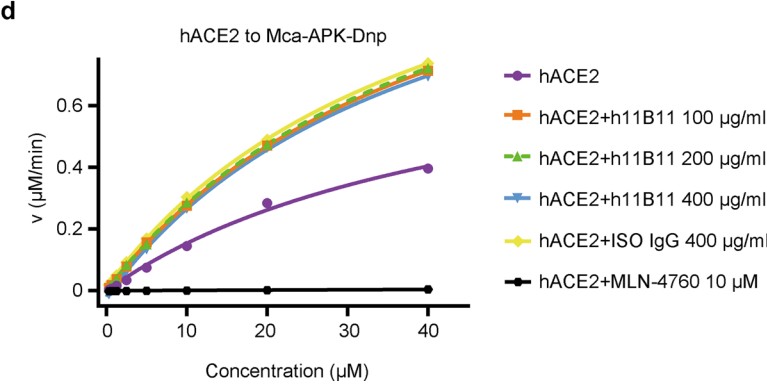

hACE2 treated with different concentrations of hACE2-blocking MAb (from 100 to 400 μg/ml) was not dramatically changed (Fig. 1d and Supplementary Table 1). Even at 400 μg/ml, a comparable $K_m$ was obtained from h11B11 and isotype IgG protein-mediated reactions (Fig. 1d and Supplementary Table 1). Moreover, the $K_{cat}/K_m$ value of hACE2 was consistent in the presence of hACE2-blocking MAb or isotype IgG (Fig. 1d and Supplementary Table 1). These data indicate that h11B11 does not affect the carboxypeptidase activity of hACE2 and that the virus binding sites and enzyme active site on hACE2 do not overlap[21], so the h11B11 MAb may not cause serious side effects in clinical treatment.

**Fig. 1 Characterization of hACE2-blocking MAb h11B11. a** h11B11 blocks SARS-CoV-RBD and SARS-CoV-2-RBD binding to hACE2. The quantified data were shown in the right panel. Mean values of three biological replicates ± SEM (standard error of the mean) are shown (unpaired *t* test, two tailed). Data are representative of three independent experiments. **b**, **c** The binding kinetics of h11B11 to wild-type and mutated hACE2 was assessed using a single-cycle model. The kinetic parameters were labeled accordingly. Values represent mean ± SEM of three independent assessments. n.d. means the values are not detectable due to poor binding ability. **d** Measurement of enzymatic kinetic constants of hACE2 in the presence of h11B11 and control IgG protein. Actual data points and Michaelis–Menten plots for hACE2 hydrolysis of Mca-AFK-Dnp in the absence (purple) or presence (orange, green, and blue) of h11B11 protein. Isotype IgG (yellow) and inhibitor (black) were used as control. The initial velocity conditions were limited to 12 min. The value in the x-axis of **d** represents the concentrations of substrate (Mca-AFK-Dnp).

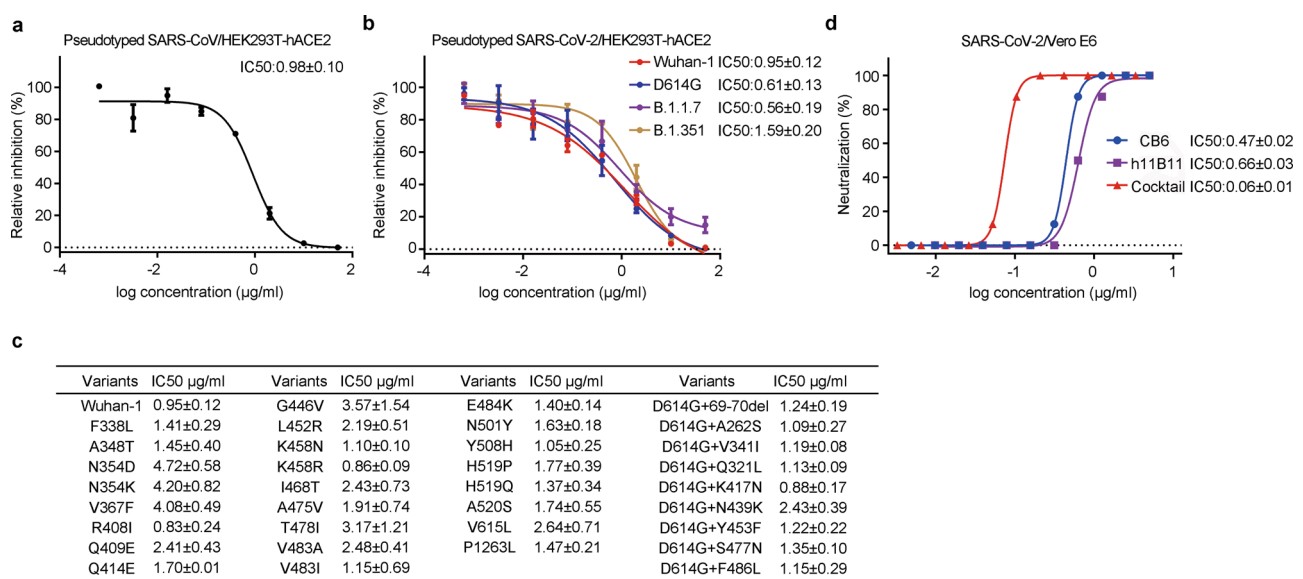

**c**

| Variants | IC50 µg/ml | Variants | IC50 µg/ml | Variants | IC50 µg/ml | Variants | IC50 µg/ml |
|---|---|---|---|---|---|---|---|
| Wuhan-1 | 0.95±0.12 | G446V | 3.57±1.54 | E484K | 1.40±0.14 | D614G+69-70del | 1.24±0.19 |
| F338L | 1.41±0.29 | L452R | 2.19±0.51 | N501Y | 1.63±0.18 | D614G+A262S | 1.09±0.27 |
| A348T | 1.45±0.40 | K458N | 1.10±0.10 | Y508H | 1.05±0.25 | D614G+V341I | 1.19±0.08 |
| N354D | 4.72±0.58 | K458R | 0.86±0.09 | H519P | 1.77±0.39 | D614G+Q321L | 1.13±0.09 |
| N354K | 4.20±0.82 | I468T | 2.43±0.73 | H519Q | 1.37±0.34 | D614G+K417N | 0.88±0.17 |
| V367F | 4.08±0.49 | A475V | 1.91±0.74 | A520S | 1.74±0.55 | D614G+N439K | 2.43±0.39 |
| R408I | 0.83±0.24 | T478I | 3.17±1.21 | V615L | 2.64±0.71 | D614G+Y453F | 1.22±0.22 |
| Q409E | 2.41±0.43 | V483A | 2.48±0.41 | P1263L | 1.47±0.21 | D614G+S477N | 1.35±0.10 |
| Q414E | 1.70±0.01 | V483I | 1.15±0.69 | | | D614G+F486L | 1.15±0.29 |

**Fig. 2 h11B11 effectively inhibits original coronaviruses and novel lineages of SARS-CoV-2 in vitro. a**, **b** Inhibiting potencies of MAb to original SARS-CoV, SARS-CoV-2, or epidemic SARS-CoV-2 variant pseudovirus were evaluated in a luciferase reporter assay. IC50 was calculated by fitting the relative fluorescence intensity of host cells from serially diluted antibodies to a sigmoidal dose–response curve. **c** h11B11 was tested against 34 pseudotyped SARS-CoV-2 variants emanated from GISAID. **d** IC50 to live SARS-CoV-2 was fit the CPE from serially diluted antibody to a sigmoidal dose–response curve after 3 days of incubation. The values in the x-axis of **a**, **b**, and **d** represent concentrations of the antibody. IC50 in (**a**–**d**) represents mean ± SEM (standard error of the mean) of triplicated independent assessments.

**MAb h11B11 broadly neutralizes SARS-CoV-2 viruses and various epidemic lineages.** To investigate how broad the range of inhibitory activity against CoVs is for h11B11, the performance of MAb blocking in pseudotyped and authentic virus-infected host cell line assays was determined in vitro. Thirty-nine SARS-CoV and SARS-CoV-2 S protein-bearing vesicular stomatitis virus (VSV delta G/luciferase) pseudoviruses were constructed for assessment. These S coding sequences include the original SARS-CoV-2, SARS-CoV-2 sustained transmission lineages, and high-frequency mutants from Global Initiative on Sharing All Influenza Data (GISAID). The median inhibitory concentration (IC50) of h11B11 in blocking SARS-CoV pseudovirus infection of HEK293T-hACE2 cells was 0.98 µg/ml, consistent with the efficacy (0.95 µg/ml) against pseudotyped SARS-CoV-2 virus (Fig. 2a, b). Notably, h11B11 exhibited strong inhibitory activity in terms of IC50 (0.61, 0.56, and 1.59 µg/ml) in all three pseudoviruses expressing the SARS-CoV-2 S antigens of lineages with sustained transmission (D614G, B.1.1.7, and B.1.351)[23,24] (Fig. 2b). Next, we set out to investigate the inhibitory activity of h11B11 against 34 pseudoviruses, including high-frequency SARS-CoV-2 variants and combined variants with D614G, across the entire S protein. Notably, some changes in the RBD region, A475V, E484K, and N501Y, were demonstrated to alter the sensitivity to SARS-CoV-2-RBD-specific neutralizing MAbs[25]. As shown in Fig. 2c, the IC50 titers of h11B11 to a single amino acid change and the double amino acid-mutated SARS-CoV-2 pseudoviruses were summarized. Although the inhibitory

activity of h11B11 was slightly impaired (~4-fold) by N354D, N354K, and V367F, its potency against SARS-CoV-2 variants was not markedly reduced.

Herein, we assessed the neutralizing activity of h11B11 against authentic viruses. The antiviral activity of single h11B11 and a combination of h11B11/CB6, a clinical neutralizing MAb, against SARS-CoV-2 was assessed in Vero E6 cells. In parallel, the mixture of h11B11 and LLC-MK2 cells was infected with HCoV-NL63 virus in a 96-well plate. After being co-incubated for 3 days, virus-induced cytopathic effect (CPE) was inspected. Consistently, h11B11 potently neutralize live SARS-CoV-2 infection of host cells in a dose-dependent inhibition profile (IC50 = 0.66 ± 0.03 µg/ml) (Fig. 2d). Importantly, the combination treatment of h11B11 and CB6 against live SARS-CoV-2 infection of cells, with an observed subnanomolar median inhibitory concentration (IC50 of 0.06 ± 0.01 µg/ml, ~0.4 nM), is one order of magnitude more potent over the single agents (Fig. 2d). In the concentration of 100 µg/ml, no inhibiting activity of the MAb to authentic HCoV-NL63 was observed (Supplementary Fig. 3). In summary, MAb h11B11 exhibits broadly substantial blocking activities against SARS-CoV, SARS-CoV-2, and various epidemic lineage infections in vitro, while the synergistic antibody cocktail exhibits superior inhibiting activity.

**h11B11 protects hACE2 mice from SARS-CoV-2 infections.** To evaluate the efficiency of h11B11 in protecting ACE2 humanized mice from SARS-CoV-2 infection, we initially assessed the impact

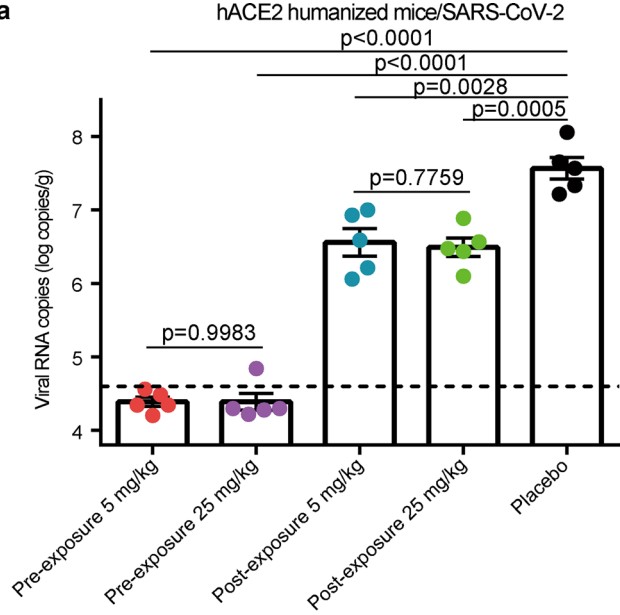

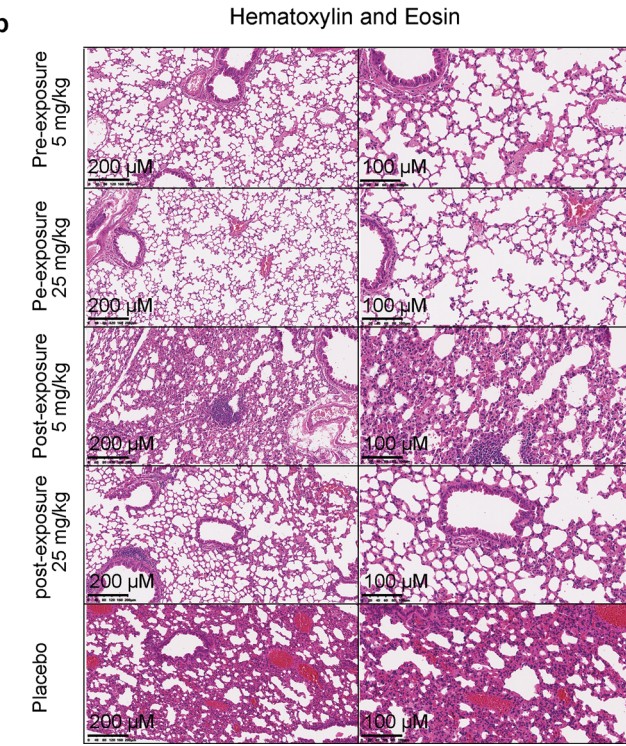

**Fig. 3 The effective protection of h11B11 in ACE2 humanized mice challenged with SARS-CoV-2. a** Groups of hACE2 mice that infected SARS-CoV-2 were divided into pre-exposure, postexposure, and control groups with eight animals in each group. Before the challenge, the animals of prophylactic (one day before infection) and treatment settings (one day after challenge) were intraperitoneally infused with h11B11. Mice in the control group were given PBS as a control. SARS-CoV-2 titers in the lungs were measured by qRT-PCR with two replicates and values represent mean ± SEM (standard error of the mean) ($n = 5$) (unpaired $t$ test, two tailed). **b** Histopathological analysis of lung samples at 5 d.p.i. ($n = 3$). Data are one representative result of two independent experiments. Scale bar: 100 and 200 µm.

of the MAb in both prophylactic and treatment settings. Homozygous mice expressing hACE2 were generated by using CRISPR/Cas9 to knock the full cDNAs of hACE2 into exon 2, the first coding exon, of the m*Ace2* gene located at GRC m38.p6[26]. Groups of mice (eight in each group) were intraperitoneally administered 5 or 25 mg/kg antibody as prophylaxis and challenged with $5 \times 10^5$ 50% tissue culture infectious dose (TCID50) of the virus through the intranasal route one day after MAb dosing. In the treatment settings, animals were treated with 5 or 25 mg/kg antibody one day after challenge with the same dosage of SARS-CoV-2 virus. Eight mice in the placebo group were synchronously infected. Owing to the relatively transient nature of the virus infection in this model, all mice were sacrificed on day 5 after the challenge. Then, the number of SARS-CoV-2 RNA copies was measured to assess the impact of MAb on viral replication. The viral titers in the lungs of the placebo group surged to $1 \times 10^7$ RNA copies/g on the fifth day (Fig. 3a). In the therapeutic (postexposure) settings, a single dose of h11B11 after SARS-CoV-2 challenge significantly decreased the SARS-CoV-2 loads in animals, with an ~10-fold reduction (Fig. 3a). Additionally, for the prophylactic (pre-exposure) groups, both high- and low-dose administration of h11B11 resulted in viral titers in the lungs below the detection limit for nine of ten mice (Fig. 3a). The accelerated clearance and almost complete ablation of SARS-CoV-2 show that h11B11 induces a strong protective effect against virus infection in vivo. Furthermore, we performed pathology analyses of the lungs from challenged mice. Control mice showed evidence of mild interstitial pneumonia, which was characterized by inflammatory cell infiltration, alveolar septal thickening, and distinctive vascular system injury upon placebo treatment (Fig. 3b). In contrast, prophylactically treated animals showed minimal evidence of interstitial pneumonia with limited pathological features when compared with the placebo group. Similar to the mice in the placebo group, the low-dose h11B11-treated mice had visible leukocyte infiltrations and slight alveolar septal thickening (Fig. 3b). Remarkably, fewer obvious inflammatory cell infiltrations or focal hemorrhages were observed in the lung sections from mice that received 25 mg/kg h11B11. These results demonstrate that MAb h11B11 is efficacious in both prophylactic and treatment models, as measured by reduced virus replication and infection-induced pathology.

**MAb h11B11 exhibits no significant toxicity to cynomolgus monkeys at a high dose.** Antibodies can induce the internalization and degradation of the targeting protein on cell surface[27,28], so we sought to examine whether incubation of MAb h11B11 had any impact on the expressional level of ACE2 on the cell surface. HEK293T-hACE2 cells were incubated with MAb h11B11 at 37 °C to allow internalization and ACE2 protein level was examined by immunoblot and flow cytometry analysis. We found that the hACE2 protein level remained unchanged on the cell membrane after incubation with MAb h11B11 at a high concentration of 10 µg/ml (Fig. 4a, b). This result was further confirmed by flow cytometry analysis that hACE2 level on the cell surface was consistent in 4 or 24 h after incubation with a series of h11B11 protein (Fig. 4c, d). Therefore, the levels of membrane-expressed hACE2 were not affected after incubating with h11B11, which indicates the safety of this MAb.

To assay the safety of h11B11 in vivo, cynomolgus monkeys were employed to explore the toxicity and the maximum-tolerated dose (MTD). An SPR assay confirmed that this MAb shows analogous binding affinities to hACE2 and cynomolgus ACE2 (Supplementary Fig. 4). Four male cynomolgus monkeys (animal No. 1001, 1002, 1101, and 1102) were randomly divided into two groups according to body weight. Cynomolgus monkeys

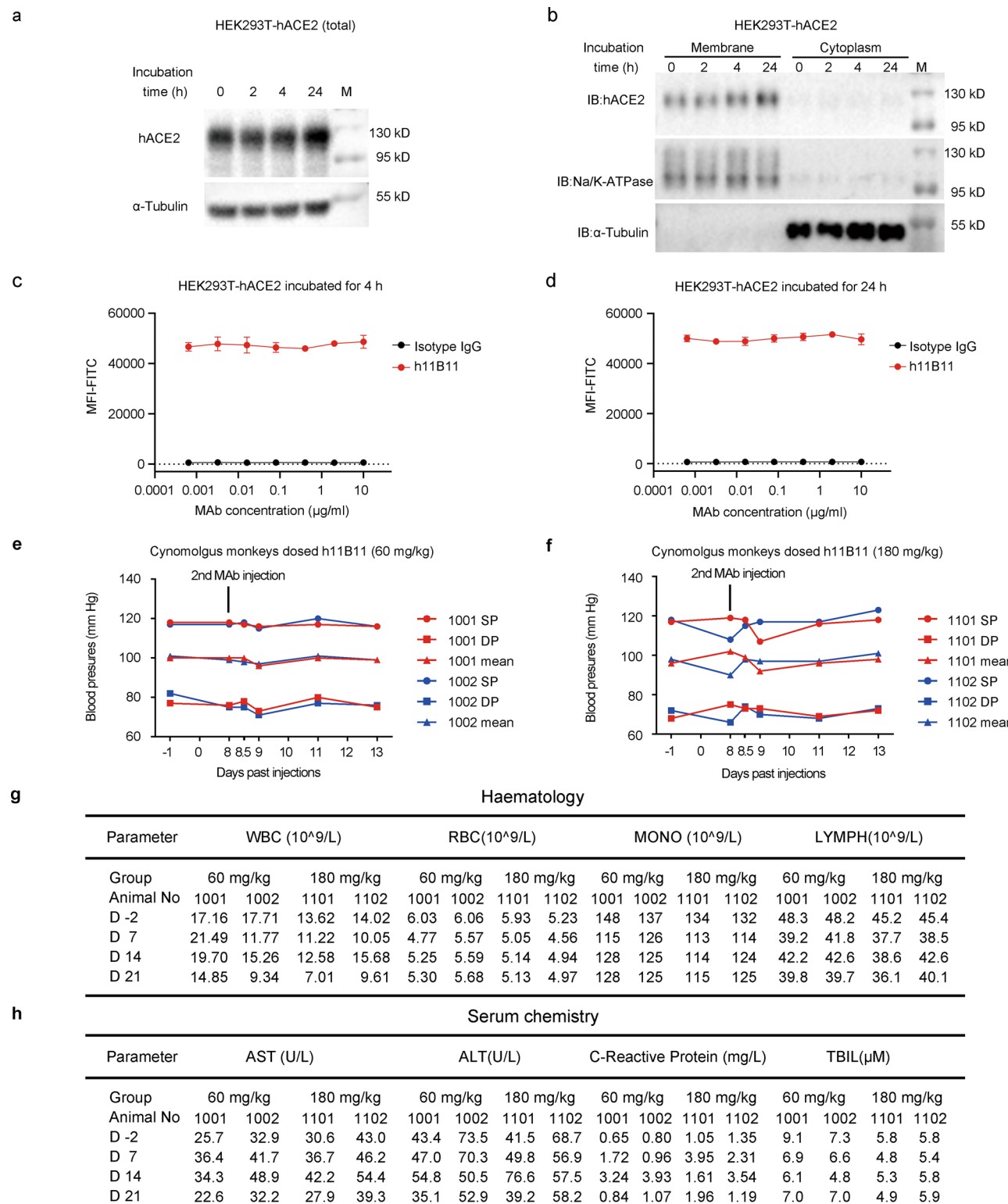

g Haematology

| Parameter | WBC (10^9/L) | | | | RBC(10^9/L) | | | | MONO (10^9/L) | | | | LYMPH(10^9/L) | | | |
|---|---|---|---|---|---|---|---|---|---|---|---|---|---|---|---|---|
| Group | 60 mg/kg | | 180 mg/kg | | 60 mg/kg | | 180 mg/kg | | 60 mg/kg | | 180 mg/kg | | 60 mg/kg | | 180 mg/kg | |
| Animal No | 1001 | 1002 | 1101 | 1102 | 1001 | 1002 | 1101 | 1102 | 1001 | 1002 | 1101 | 1102 | 1001 | 1002 | 1101 | 1102 |
| D -2 | 17.16 | 17.71 | 13.62 | 14.02 | 6.03 | 6.06 | 5.93 | 5.23 | 148 | 137 | 134 | 132 | 48.3 | 48.2 | 45.2 | 45.4 |
| D 7 | 21.49 | 11.77 | 11.22 | 10.05 | 4.77 | 5.57 | 5.05 | 4.56 | 115 | 126 | 113 | 114 | 39.2 | 41.8 | 37.7 | 38.5 |
| D 14 | 19.70 | 15.26 | 12.58 | 15.68 | 5.25 | 5.59 | 5.14 | 4.94 | 128 | 125 | 114 | 124 | 42.2 | 42.6 | 38.6 | 42.6 |
| D 21 | 14.85 | 9.34 | 7.01 | 9.61 | 5.30 | 5.68 | 5.13 | 4.97 | 128 | 125 | 115 | 125 | 39.8 | 39.7 | 36.1 | 40.1 |

h Serum chemistry

| Parameter | AST (U/L) | | | | ALT(U/L) | | | | C-Reactive Protein (mg/L) | | | | TBIL(μM) | | | |
|---|---|---|---|---|---|---|---|---|---|---|---|---|---|---|---|---|
| Group | 60 mg/kg | | 180 mg/kg | | 60 mg/kg | | 180 mg/kg | | 60 mg/kg | | 180 mg/kg | | 60 mg/kg | | 180 mg/kg | |
| Animal No | 1001 | 1002 | 1101 | 1102 | 1001 | 1002 | 1101 | 1102 | 1001 | 1002 | 1101 | 1102 | 1001 | 1002 | 1101 | 1102 |
| D -2 | 25.7 | 32.9 | 30.6 | 43.0 | 43.4 | 73.5 | 41.5 | 68.7 | 0.65 | 0.80 | 1.05 | 1.35 | 9.1 | 7.3 | 5.8 | 5.8 |
| D 7 | 36.4 | 41.7 | 36.7 | 46.2 | 47.0 | 70.3 | 49.8 | 56.9 | 1.72 | 0.96 | 3.95 | 2.31 | 6.9 | 6.6 | 4.8 | 5.4 |
| D 14 | 34.3 | 48.9 | 42.2 | 54.4 | 54.8 | 50.5 | 76.6 | 57.5 | 3.24 | 3.93 | 1.61 | 3.54 | 6.1 | 4.8 | 5.3 | 5.8 |
| D 21 | 22.6 | 32.2 | 27.9 | 39.3 | 35.1 | 52.9 | 39.2 | 58.2 | 0.84 | 1.07 | 1.96 | 1.19 | 7.0 | 7.0 | 4.9 | 5.9 |

in the high-dose group (180 mg/kg to 1101 and 1102) or the low-dose group (60 mg/kg to 1001 and 1002) were administered via repeated intravenous infusion (once a week for 3 weeks) at a dosage of 1 ml/kg at a rate of 1 ml/min. Evaluation indicators included blood pressure and clinical pathology (hematology and serum biochemistry). During the study, the animals in each group survived until the planned euthanasia. To enable animals to adapt to the vest used for measuring blood pressure and reflect the situation after multiple administrations, we monitored the changes in blood pressure after the second administration. Compared with the predose measurements, the mean diastolic and systolic blood pressure of all animals in the following 6 days after the second dosing showed only minor changes, from 90 to 102 mm Hg (Fig. 4e, f). The fluctuation of blood pressure in the low-dose group was smaller (from 96 to 101 mm Hg) (Fig. 4e) than that of high-dose injected animals (from 90 to 102 mm Hg) (Fig. 4f). Meanwhile, the hematology assays showed that the white blood cell (WBC), red blood cell (RBC), monocyte, and

**Fig. 4 The safety of h11B11 in vitro and in cynomolgus monkeys. a** HEK293-hACE2 cells were incubated with 10 μg/ml h11B11 antibody for the indicated times, followed by western blot analysis of indicated proteins. **b** HEK293-hACE2 cells were incubated with 10 μg/ml h11B11 antibody for the indicated times, followed by membrane–cytoplasm fractionation analysis of indicated proteins. The antibody used for western blot in **a**, **b** was anti-ACE2 pAb. Data are one representative result of two independent experiments. **c** HEK293-hACE2 cells were incubated with serially diluted h11B11 antibody or anti-IgG control antibody. After incubation at 37 °C for 4 h (**c**) or 24 h (**d**), the hACE2–antibody complex was detected by flow cytometry. Mean values of three biological replicates mean ± SEM (standard error of the mean) are shown. Data are representative of three independent experiments. **e**, **f** The blood pressure of cynomolgus monkeys in low and high groups after the second dosing. 1001, 1002, 1101, and 1102 represent animal numbers. SP is the systolic pressure and DP is the diastolic pressure. Mean is the average of SP and DP. The value was tested once. **g**, **h** The hematology and serum chemistry (toxicology) assays of four animals for 3 weeks. Cynomolgus monkeys were injected 60 or 180 mg/kg h11B11 once a week for three times. WBC white blood cell, RBC red blood cell, MONO monocyte, LYMPH lymphocyte, AST aspartate transaminase, ALT alanine transaminase, TBIL total bilirubin.

lymphocyte counts of all cynomolgus monkeys were equally stable at 21 days (Fig. 4g). To evaluate the toxicity of h11B11 to animals, aspartate transaminase, alanine transaminase, C-reactive protein, and total bilirubin were assayed. These indexes of the animals in the two groups were not significantly affected by multiple administrations (Fig. 4h). While histopathology tissue slides and microscopy observation were not conducted, multiple dosages of h11B11 were observed to be safe in this preclinical setup, and the MTD was 180 mg/kg.

**Analysis of the MAb blocking mechanism**. The binding characteristics of h11B11 to hACE2 were subsequently explored through the determination of the complex structure. For crystal screening, the extracellular domains of hACE2 and h11B11-Fab proteins were prepared in a eukaryotic cell expression system. The structure of the h11B11/hACE2 complex was determined at a resolution of 3.8 Å (Supplementary Table 2). The overall structure reveals that h11B11 binds to hACE2 with a buried surface area of 266 Å$^2$. For further analysis of possible interactions, a cutoff value of 4.5 Å is used to define the main contacts between h11B11 and hACE2. All three CDRs of the VH and LCDR1 and LCDR3 provide contacts with hACE2, while LCDR2 is not engaged (Fig. 5a). The complex indicates that the binding of h11B11 is mainly located on the NTH of hACE2.

To analyze the mechanism of hACE2 antagonism of the blocking MAb, the structure of the h11B11-Fab/hACE2 complex was superimposed onto the SARS-CoV-RBD/hACE2 (Protein Data Bank (PDB) ID: 2AJF), SARS-CoV-2-RBD/hACE2 (PDB ID: 6LZG), and HCoV-NL63-RBD/hACE2 complexes (PDB ID: 3KBH). Detailed comparisons of the reciprocal binding areas between these complexes revealed many striking parallels. Three RBDs bind similarly to the NTH of hACE2. Specifically, the binding of h11B11 to hACE2 displayed substantial hindrance to that of SARS-CoV-RBD and SARS-CoV-2-RBD (Fig. 5b, d, e), but not to that of HCoV-NL63-RBD (Fig. 5b, c). Further analysis of the binding surface of h11B11 on hACE2, in comparison to that of SARSr-CoV-RBDs, revealed that the overlapping binding surface is mainly located on the N-terminal amino acids of NTH (Fig. 5d, e). In contrast, the NTH region of hACE2 that interacts with MAb h11B11 is also not involved in competitive binding to HCoV-NL63-RBD (Fig. 5c). Therefore, h11b11 blocks SARS-CoV-RBD and SARS-CoV-2-RBD to bind to hACE2 mainly through N-terminal epitope competition and steric clash. No structural evidence directly indicates that the MAb h11B11 inhibits the binding of HCoV-NL63-RBD and ACE2.

## Discussion

As a decoy, recombinant soluble ACE2 or ACE2-Fc fusion protein is a therapeutic countermeasure for use against COVID-19, but the specificity and affinity of wild-type ACE2 are usually lower than those of the ACE2-blocking MAb h11B11[2,29]. Although affinity-enhancing mutations provide ACE2 variants

with affinities that rival those of monoclonal antibodies, foreign amino acid substitutions induce the potential risk of immunogenicity and reduce the proteolytic activity of angiotensin II, which is a second therapeutic option to relieve symptoms of respiratory distress[30].

During the revision of this article, dalbavancin, which has been approved for the treatment of acute bacterial skin and skin structure infections caused by designated susceptible gram-positive bacteria in adults, was reported to prevent SARS-CoV-2 infection in animal models[31]. In the preventive group, the viral load changes of dalbavancin-treating mice were significantly weaker than those of MAb h11B11 (Fig. 3a). Combined with our results, these reports suggest that antibodies are more potent than dalbavancin as countermeasures against SARS-CoV-2.

Since the outbreak of the COVID-19 pandemic, several groups have isolated human MAbs from B cells of convalescing individuals that bind to the SARS-CoV-2 S glycoprotein and exhibit potent neutralizing activity to SARS-CoV-2[4,5,32–37]. In both prophylactic and therapeutic treatments, MAb cocktails from two antigenic sites on SARS-CoV-2-RBD were shown to synergize and protect against infections in animal models[5] and clinical trials[38,39], yet extensive mutations in the SARS-CoV-2 S protein could lead to antigenic changes resistant to antibody therapies[8,9]. To 37 naturally occurring pseudotyped SARS-CoV-2 variants, h11B11 exhibits potent inhibitory activity. Moreover, the recent emergence of new SARS-CoV-2 variants (D614G, B.1.1.7, and B.1.351) was reported to be easily transmitted and refractory to neutralization by some RBD-specific MAbs[8] that are not resistant to h11B11 (Fig. 2b, c). It is not surprising that promising molecules with strong activity also block SARS-CoV from recognizing their entry receptors. Notably, the structural comparisons again highlighted the evidence of steric hindrance and epitope competition between h11B11 and SARS-CoV-RBD or SARS-CoV-2-RBD to clarify the blocking mechanism. Meanwhile, the overall structure corroborated the relationship of no inhibitory activity and no structural conflict of HCoV-NL63 and h11B11.

The COVID-19 pandemic has been reported in 223 countries and regions worldwide (World Health Organization). In different racial subpopulations, ACE2 polymorphisms were likely associated with the genetic susceptibility to COVID-19[20]. We found that a low-frequency locus, S19P (rs73635825, $p = 7.9 \times 10^{-4}$), in the 1KGP (1000 Genomes Project) database abolished h11B11 binding to this ACE2 variant (Fig. 2c). In an analysis of its incidence, no distribution of this nominal variant was found, except in an African population ($p = 3 \times 10^{-3}$)[20]. These findings suggested that the ACE2-blocking MAb could play an active role in broad populations. Meanwhile, we speculate two reasons for the S19P variant eliminating the interaction between hACE2 and MAb h11B11. Although multiple contacts co-influence the binding affinity, a single key site mutation could ablate the binding between MAbs and antigens. For example, E484K on SARS-CoV-2-RBD completely abolished or substantially reduced the activities of MAb 2-15 and C121[40]. Another possibility is that

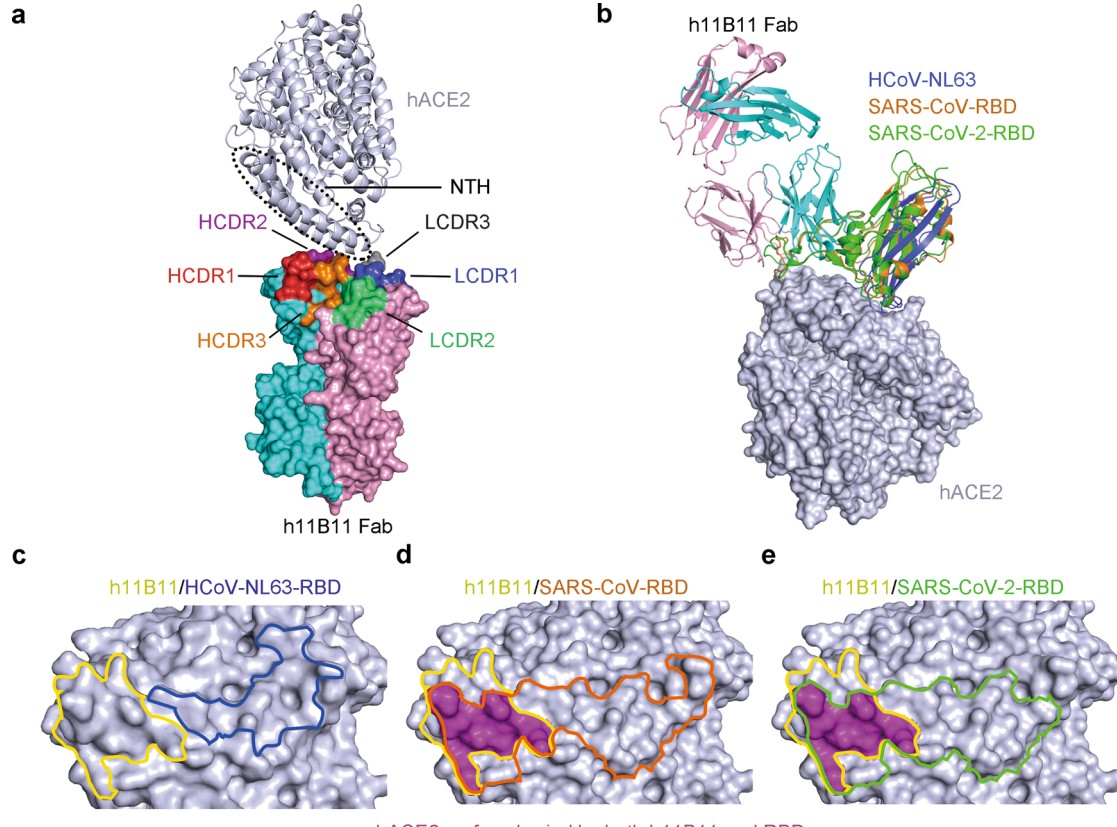

**Fig. 5 Structural basis of the competitive binding of h11B11 and RBDs with hACE2. a** The overall structure of h11B11-Fab and hACE2. The Fab is shown as the surface with HCDR1, HCDR2, and HCDR3 from the heavy chain (cyan) colored in red, orange, and purple, while the LCDR1, LCDR2, and LCDR3 from light chain (pink) are colored in blue, green, and gray, respectively. The hACE2 is shown as a cartoon. The N-terminal helix (NTH) is marked by a dotted line. **b** Superimposition of h11B11/hACE2 complex and HCoV-NL63-RBD/hACE2 (PDB:3KBH), SARS-CoV-RBD/hACE2 (PDB:2AJF), and SARS-CoV-RBD/hACE2 (PDB:6LZG) reveal the competition between h11B11 and RBDs. The ACE2, HCoV-NL63-RBD, SARS-CoV-RBD, and SARS-CoV-RBD are colored differently as indicated. **c–e** Competitive binding surfaces of h11B11 and RBDs on hACE2. The overlapping epitopes bound by both h11B11 and RBDs are colored in purple. The surfaces in contact with MAb are colored in yellow, while others in contact with RBDs are colored differently as indicated.

proline substitution could destroy the conformation of the NTH[41], which leads to the disappearance of the interaction.

Completed clinical trials have demonstrated that active ACE2 can serve as a potential therapy to alleviate pulmonary injury, vascular damage, and lung fibrosis[21]. On the basis of similarities with these pathological characteristics, the h11B11-promoted carboxypeptidase activity of the ACE2 receptor synergistically relieves symptoms of COVID-19. Importantly, due to the long half-life of antibody drugs[42], clinical trials of antibody drugs related to COVID-19 have adopted a single administration research scheme, and the MAb dosages have been <50 mg/kg[38,39]. Due to the increasingly limited animal resources, four male cynomolgus monkeys were employed in the preliminary toxicity study of antibody h11B11. In our study, injections of three doses of 180 mg/kg h11B11 did not induce drastic blood pressure changes or obvious toxicological damage in nonhuman primate models. Considering the limited animal numbers and the lack of histopathology assays, a comprehensive GLP-compliant safety evaluation is needed.

In conclusion, our work shows that the ACE2-blocking MAb represents a broadly promising therapeutic candidate against emergence, re-emergence, and future zoonotic transmission events from SARSr-CoVs and variants.

## Methods
**Cell lines and viruses**. HEK293T (ATCC, CRL-3216), HEK293T-ACE2 (Sino-Biological, OEC001), Vero E6 (ATCC, CRL-1586), and LLC-MK2 (ATCC, CRL-7)

cells were cultured at 37 °C under 5% $CO_2$ in Dulbecco's modified Eagle's medium (DMEM) (HyClone, South Logan, UT) supplemented with 10% fetal bovine serum (FBS) (Gibco, Carlsbad, CA, USA).

SARS-CoV-2 virus (BetaCoV/Wuhan/IVDC-HB-01/2020, GISAID accession ID: EPI_ISL_402119) was isolated by National Institute for Virus Disease Control and Prevention, Chinese Center for Disease Control and Prevention. Vero E6 cells were applied to the reproduction of SARS-CoV-2 stocks. The HCoV-NL63 strain was isolated by the Chinese Academy of Medical Sciences and Peking Union Medical College. LLC-MK2 cells were applied to the reproduction of HCoV-NL63 stocks.

**Plasmid construction**. The coding sequences of SARS-CoV-RBD (residues 306–527, accession number: NC_004718), SARS-CoV-2-RBD (residues 319–541, accession number EPI_ISL_402119), hACE2 (residues 19–615, accession number BAJ21180), and hACE2 variants (S19P, I21T, K26R, N33D, and D38E) fused with N-terminal native signal peptides and C-terminal 6× His tag were, respectively, cloned into the pCAGGS expression vector (Addgene) using the *Eco*RI and *Xho*I restriction sites. The signal peptides and variable regions of h11B11 were synthesized (GenScript) and fused with the coding sequences for the human IgG$_4$ and kappa light chain constant region into the pCAGGS vectors. The pEGFP-N1-hACE2 plasmid was constructed by cloning the coding region of hACE2 into pEGFP-N1 using restriction enzymes *Xho*I and *Sma*I. To express minimal glycosylated ACE2, a coding sequence of residues 19–615 was synthesized (GenScript) and cloned into pFastBac1 vector (Invitrogen), with an N-terminal gp67 signal peptide and a C-terminal 6×His tag.

**Protein expression and purification**. To prepare the proteins of ACE2 (19–615), SARS-CoV-RBD, and SARS-CoV-2-RBD, HEK293T cells were transiently transfected with expressing plasmids containing the coding sequence for the indicated proteins. After 3 days, the supernatant was collected and soluble protein was purified by Ni affinity chromatography using a HisTrap HP 5 ml column (GE Healthcare). The samples were then further purified via size-exclusion

chromatography with a Superdex 200 column (GE Healthcare) in a buffer composed of 20 mM Tris-HCl (pH 8.0) and 150 mM NaCl. Preparation of the full-length h11B11 was achieved by transfection of plasmids into HEK293T cells. The protein was purified from the culture supernatants using a HiTrap Protein A HP column (GE Healthcare) and subsequently purified via the above size-exclusion chromatography.

For crystal screenings, the peptidase domain of human ACE2 (19–615) with a C-terminal 6×His tag was expressed using the baculovirus–insect cell system. The baculovirus was generated and amplified using the Sf21 insect cells (Invitrogen, B82101), and Hi5 insect cells (Invitrogen, B85502) were used for protein expression. The conditioned medium was collected 48 h post infection and exchanged into the binding buffer (10 mM HEPES, pH 7.2, and 150 mM NaCl). The ACE2 (19–615) and h11B11-Fab proteins were purified as described above for HEK293T cell-derived ACE2 (19–615). To obtain the complex between ACE2 and h11B11-Fab, purified ACE2 and h11B11-Fab were incubated together, passed through a Superdex 200 increase 10/300 gel filtration column (GE Healthcare), and eluted using the binding buffer.

**Flow cytometry assay.** To test the activity of antibodies to block the binding between ACE2 and SARS-CoV-RBD, or SARS-CoV-2-RBD. HEK293T cells were transiently transfected with pEGFP-N1-ACE2 plasmids. After 24 h, $3 \times 10^5$ cells were collected and incubated with 10 μg/ml h11B11 protein or isotype IgG at 37 °C for 30 min, followed by incubation with 200 ng/ml RBD proteins at 37 °C for another 30 min. After washing three times, the cells were incubated with APC-conjugated anti-His antibody (1:200, Miltenyi Biotec, 130-119-782) for another 30 min. Then, the cells and data were collected and analyzed using flow cytometry (BD FACS Canto™ II, BD FACSDiva Software v8.0.3, and FlowJo 7.6.1). The gating strategy was provided in Supplementary Fig. 5.

To test whether the h11B11 antibody has any impact on the cell-surface expression of hACE2, HEK293T-hACE2 cells were incubated with different concentrations (10 μg/ml or with five-fold serial dilutions ranging from 10 μg/ml to 0.64 ng/ml) of h11B11 at 37 °C in DMEM with 10% FBS for 4 or 24 h. Then, the cells were washed with FACS buffer (phosphate-buffered saline (PBS), 1% bovine serum albumin, and 2 mM EDTA) and incubated with 10 μg/ml h11B11 antibody or isotype IgG at 4 °C for 60 min. After washing three times, cells were incubated with Alexa Fluor™488 goat anti-human IgG (H + L) antibody (1:200, Invitrogen, A11013) at 4 °C for another 30 min. Then, the cells were washed twice and resuspended in 200 μl FACS buffer for flow cytometry analysis (Beckman CytoFLEX S, Beckman CytExpert 2.3.0.84, and FlowJo 7.6.1). The gating strategy was provided in Supplementary Fig. 6.

**Surface plasmon resonance.** The interaction between h11B11 with hACE2 was monitored by SPR using a BIAcore 8K (GE Healthcare) carried out in single-cycle mode with protein A biosensor chip (GE Healthcare). All the measurements were performed in the buffer consisting of 10 mM $Na_2HPO_4$, 2 mM $KH_2PO_4$, 137 mM NaCl, 2.7 mM KCl, pH 7.4, and 0.05% (v/v) Tween-20. The antibody protein was captured on the chip at ~1000 response units. Then, gradient concentrations of ACE2 protein (from 200 to 12.5 nM with two-fold dilutions) flowed over the chip surface and the real-time response was recorded. After each cycle, the sensor was regenerated with 10 mM Gly-HCl (pH 1.5). The raw data and affinities were collected and calculated using a 1:1 fitting model with BIAevaluation software (GE Healthcare, Biacore 8 K Control Software 2.0.15.12933 and Biacore Insight Evaluation 1.0.5.11069).

**hACE2 carboxypeptidase activity measurement.** Enzymatic reactions were performed in black microtiter plates at ambient temperature (26 °C). To each well, 25 μl of 1.6 μg/ml hACE2 (19–615) protein in PBS was added, respectively. Then, 25 μl h11B11 proteins at various final concentrations of 100, 200, and 400 μg/ml or hACE2 inhibitor (MLN-4760, Sigma, 5.30616) at a final concentration of 10 μM were added to wells and incubated for 15 min. The reactions were initiated by adding 50 μl of fluorogenic peptides (Mac-APK-Dnp) (GenScript) at 40 μM or with two-fold serial dilutions ranging from 40 to 0.3125 μM to determine the kinetic constants for hACE2 hydrolysis. The relative fluorescence units (RFUs) were read at excitation and emission wavelengths of 320 and 405 nm, respectively, in kinetic mode at 2-min intervals for 6 h (BMG LABTECH, CLARIOstar Plus 5.61). To calculate the specific activity of hACE2, the intensities of RFU were converted to molarities according to standard substrate Mca-P-L-OH (GenScript). To obtain the kinetic constants, the initial velocity conditions were limited to 12 min. Initial velocities were plotted versus substrate concentration and fit to the Michaelis–Menten equation $v = V_{max}[S]/(K_m + [S])$ using GraphPad Prism software (version 6.0). Turnover numbers ($k_{cat}$) were calculated from the equation $k_{cat} = V_{max}/[E]$, using the hACE2 molecular mass of 85 kDa and assuming the enzyme sample to be essentially pure and fully active.

**Generation of pseudoviruses.** pcDNA3.1.S2 recombinant plasmid (GenBank: MT_613044), constructed by inserting the codon-optimized S gene of SARS-CoV-2 (GenBank: MN_908947) into pcDNA3.1[25], was used as the template to generate the plasmid with mutagenesis in S gene. Following the procedure of circular PCR, 15–20 nucleotides before and after the target mutation site were selected as forward

primers, while the reverse complementary sequences were selected as reverse primers. Following site-directed mutagenesis PCR, the template chain was digested using *Dpn*I restriction endonuclease (NEB, R0176S). Afterward, the PCR product was directly used to transform *Escherichia coli* DH5α-competent cells (Vazyme, C502-02) and single clones were selected and then sequenced.

The SARS-CoV and SARS-CoV-2 pseudoviruses were produced using the VSV pseudovirus system as described previously[43]. In brief, on the day before transfection, HEK293T cells were prepared and adjusted to the concentration of $5 \times 10^5$ cell/ml, 15 ml of which were transferred into a T75 cell culture flask and incubated overnight at 37 °C in an incubator conditioned with 5% $CO_2$. The cells generally reach 70–90% confluence after overnight incubation. Thirty micrograms of DNA plasmid expressing the spike protein was transfected according to the user's instruction manual of Lipofectamine 3000 (Invitrogen, L3000001). The transfected cells were subsequently infected with G∗ΔG-VSV (VSV G-pseudotyped virus) (from Professor Michael A. Whitt of the University of Tennessee, USA) at concentrations of $7 \times 10^5$ TCID50/ml. After being incubated for 6 h, the medium was replaced with a fresh medium and incubated for 24 h. The culture supernatants containing the pseudovirus were harvested, filtered (0.45 μM pore size), and stored at −80 °C. TCID50 of pseudoviruses was determined as described previously[43].

**Neutralization assay.** For pseudovirus neutralization assay, $10^4$ HEK293T-hACE2 cells per well were seeded into 96-well plates (Corning) before infection. Fifty-five microliters of three- or five-fold serially diluted h11B11 protein (from 50 μg/ml) were added to cells. After incubation at 37 °C for 1 h, $1.3 \times 10^4$ TCID50 of SARS-CoV-2 pseudovirus in 55 μl were added in mixtures and subsequently incubated for 24 h. Transfer cell lysates (50 μl/well) were placed into luminometer plates (Microfluor 96-well plates). Add luciferase substrate (50 μl/well) was included in a luciferase assay system. The infectivity was determined by measuring the bioluminescence (Promega, GLoMax 1.9.3).

For live neutralization assay, $10^4$ Vero E6 cells per well were seeded in 96-well plates (Corning) before infection. Fifty microliters of two-fold serially diluted h11B11 protein (from 10 μg/ml) was added to Vero E6 cells with eight replicates. After incubation at 37 °C for 1 h, 100 TCID50 of SARS-CoV-2 in 50 μl was added to cells. In parallel, $10^4$ LLC-MK2 cells per well were seeded in 96-well plates (Corning) before infection. Fifty microliters of two-fold serially diluted h11B11 protein (from 100 μg/ml) was added to the cells with eight replicates. After incubation at 37 °C for 1 h, 20 TCID50 of HCoV-NL63 in 50 μl was added to the mixtures. Then, mixtures were subsequently incubated at 37 °C for 3 days. Cells infected with or without the virus were applied as positive or negative controls. CPE in each well was observed and recorded on the third day. A virus back titration was performed to assess the correct virus titer used in each experiment. All experiments followed the standard operating procedures (SOPs) of the approved Biosafety Level-3 facility.

**Membrane–cytoplasm fractionation assay.** HEK293T-hACE2 cells were incubated with 10 μg/ml h11B11 at 37 °C in DMEM with 10% FBS at different time points. Then, the cells were rinsed with cold 1×PBS three times and washed once with a hypotonic buffer (10 mM KCl, 1.5 mM $MgCl_2$, 10 mM Tris-HCl, pH 7.5) supplemented with a protease inhibitor. Cells were incubated on ice in the hypotonic buffer for 15 min and then pipetted up and down 5–10 times. The cell lysates were centrifuged at 4 °C for 5 min at $2500 \times g$ to remove nuclei and cellular debris. Supernatants were centrifuged at $100,000 \times g$ for 60 min at 4 °C to separate cytosolic extracts (S100) and pellets (P100). The pellets (P100) were resuspended in lysis buffer volumes equal to those of the supernatants (S100) and stored with the addition of 5× loading buffer. Western blot analysis for ACE2 and Na/K ATPase protein was carried out using rabbit anti-ACE2 pAb (1:1000, Proteintech, 21115-1-AP) and rabbit anti-Na/K ATPase (1:5000, Proteintech, 14418-1-AP). Anti-α-tubulin antibodies (1:1000) were bought from CST (2125).

**Mice experiments.** All animal experiments were carried out according to the procedures approved by the Chinese Academy of Sciences and complied with all relevant regulations regarding animal research.

hACE2 transgenic mice were described previously. Briefly, the full cDNAs of hACE2 were knocked into the exon 2, the first coding exon, of the m*Ace2* gene located in GRC m38.p6 sites. hACE2 transgenic mice (female, 30 weeks old, National Institutes for Food and Drug Control) were divided into five groups including eight mice in the placebo group injected with PBS. Animals in the pre-exposure groups were injected with 5 or 25 mg/kg antibody one day before the viral challenge. In the post-exposure groups, the mice were administered with 5 or 25 mg/kg antibody one day after the viral challenge. All mice were euthanized on the fifth day after being challenged with $5 \times 10^5$ TCID50 of SARS-CoV-2. The lung tissues from five mice in each group were placed into 1 ml of DMEM separately. After homogenization, viral RNAs were extracted by Magnetic Bead Extraction Kit (EmerTher, RE01) according to the manufacturer's instructions and eluted in 50 μl of elution buffer and used as the template for reverse transcription-polymerase chain reaction (RT-PCR). The pairs of primers were used to target *ORF1ab* gene: OFR1ab-F, 5′-CCCTGTGGGTTTTACACTTAA-3′ and OFR1ab-R, 5′-ACGATTG TGCATCAGCTGA-3′; Probe-ORF1ab 5′-the FAM-CCGTCTGCGGTATGTGG AAAGGTTATGG-BHQ1-3′. Five microliters of RNA was used to verify the RNA

quantity by One Step PrimeScript RT-PCR Kit (Takara, RR064B) according to the manufacturer's instructions. The amplification was performed as follows: 42 °C for 5 min, 95 °C for 10 s, followed by 40 cycles consisting of 95 °C for 3 s, 60 °C for 30 s, and a default melting-curve step in an Applied QuantStudio 5 Real-Time PCR System (QuantStudio Design and Analysis Software v1.5.1). The limit of detection in this RT-PCR program is 40 copies. When the detection is lower than 40 copies, the value is recorded as 20 copies.

**Histopathology and pathology**. Mice necropsies were performed according to a standard protocol. The lung tissues of three mice in each group for histological examination were stored in 10% neutral-buffered formalin for 7 days, embedded in paraffin, sectioned, and stained with hematoxylin before examination by light microscopy.

**Safety assessment**. The non-GLP safety assessment was conducted in Jiangsu Tripod Preclinical Research Laboratories Co., Ltd. All portions of this study were performed under the study protocol, any amendments (if any), and local SOPs. L.L. and J.Z. evaluated the safety assessment data and H.F. supervised the study. In addition, the nonclinical safety program was conducted following technical guidelines for repeated dose toxicity studies of pharmaceuticals issued by the National Medical Products Administration and Preclinical Safety Evaluation of Biotechnology-Derived Pharmaceuticals issued by the International Conference on Harmonization (ICH S6).

Purpose-bred cynomolgus monkeys (*Macaca fascicularis*) were obtained from licensed vendors and underwent standard quarantine periods (~4 weeks) before initiation. During the study periods, animals were single-housed in primary enclosures compliant with specifications set by the animal welfare of Jiangsu Tripod Preclinical Research Laboratories Co., Ltd. All experimental procedures (the management, sampling, and euthanasia) were conducted in Association for Assessment and Accreditation of Laboratory Animal Care (AAALAC) accredited facilities under a protocol approved by the Animal Care and Use Committee of Jiangsu Tripod Preclinical Research Laboratories Co., Ltd.

In this study, a total of four male cynomolgus monkeys (3 years old) were selected and randomly divided into two groups according to body weight. Cynomolgus monkeys were administered via repeated intravenous infusion (60 or 180 mg/kg at once a week for 3 weeks). During the study, the animals in each group survived until the planned euthanasia. At the end of the dosing period (D22), all animals were euthanized.

Clinical signs of toxicity were subjectively determined following standard procedures. Blood samples for hematology and clinical chemistry were drawn pre-study, D7, D14, and D21. Comprehensive hematology evaluations included determinations of differential leukocyte count and indicators of erythrocyte mass (RBC count). Meanwhile, serum chemistry analyses including the determination of serum enzyme activity were employed. Blood pressure measurements (systolic, diastolic, and mean blood pressure) were conducted on 6, 12, 24, 72, and 120 h after the completion of infusion on D8. Blood pressure (ecgAUTO v3.3.0.20).

According to the American Veterinary Medical Association principle, the amount of anesthetic is calculated based on the animal's body weight. At the end of the dosing period (D22), the animals were intramuscularly injected with 5 mg/kg Zoletil 50 (Virbac) combined with 2 mg/kg Sumianxin II (Dunhua Shengda Animal Co., Ltd). Anesthesia euthanasia was performed after femoral artery/venous release.

**Crystallization and structure determination**. The ACE2/h11B11-Fab complex was concentrated to 12 mg/ml for crystallization. Crystals were grown at 20 °C using the sitting-drop vapor diffusion method. The crystallization solution contains 0.1 M sodium citrate tribasic dihydrate, pH 6.5, and 30% polyethylene glycol 550. For data collection, the crystals were transferred to a solution containing the crystallization solution supplemented with 5% ethylene glycol and flash-cooled in liquid nitrogen. X-ray diffraction data were collected at the Shanghai Synchrotron Radiation Facility (beamline BL17U1). The data were collected and processed using XDS Program Package (January 31, 2020) and HKL2000 (HKL Research, v715). The structures were solved by the molecular replacement method using the Phaser program[44] in Phenix (1.10.1-2155)[45]. The closed conformation of ACE2 (PDB ID: 1R4L) and a homolog model of h11B11-Fab generated using the Phyre2 server (www.sbg.bio.ic.ac.uk/phyre2)[46] were used as search models. The structural models were then adjusted in Coot (0.8.9)[47] and refined using Rosetta refinement in Phenix (1.10.1-2155)[48]. The images of structures were generated in PyMOL (2.3.3).

**Statistical analysis**. Statistical significance between the two groups was determined by unpaired two-tailed *t* test. For the inhibition and neutralization experiments, IC50 and ND50 were calculated with the log (inhibitor) versus response–variable slope in GraphPad Prism 6.0. Enzyme kinetics ($K_m$ and $V_{max}$) of ACE2 was fit with Michaelis–Menten in GraphPad Prism 6.0. Results are shown as mean and the error bar represents standard error of the mean technical or biological as indicated in the figure legends.

**Reporting summary**. Further information on research design is available in the Nature Research Reporting Summary linked to this article.

## Data availability

The atomic coordinates and diffraction data generated in this study have been deposited in the Protein Data Bank (PDB) database under accession code 7E7E. The coding sequences of antibody h11B11 used in this study are available in the GenBank database under accession codes MZ514137 and MZ514138. The raw data generated in this study are provided in the Source Data file. Source data are provided with this paper.

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

## Acknowledgements
We thank the staff of the BL17U1 beamline at the Shanghai Synchrotron Radiation Facility (SSRF) for data collection. We are grateful to L. Ren (Institute of Pathogen Biology, Chinese Academy of Medical Sciences and Peking Union Medical College) for providing HCoV-NL63 virus. We also thank C. Fan (National Institutes for Food and Drug Control) for providing hACE2 transgenic mice. This work was supported by the Strategic Priority Research Program of CAS (XDB29040201), National Natural Science Foundation of China (81830050 and 81871280), the Junior Thousand Talents Program of China (to C.W.), and the Emergency Grant for COVID-19 of Huazhong University of Science and Technology (2020kfyXGYJ110).

## Author contributions
C.W. and J.Y. initiated and coordinated the project. R.S., Y.W., C.W., and J.Y. designed the experiments. Y.D. and R.S. sequenced and constructed the antibodies. Y.Z., Z.W., Y.D., and X.D. expressed and purified proteins. R.S. conducted the SPR and FACS analysis. X.D., Q.P., T.P., W.S., Y.D., W.H., and W.T. evaluated the neutralizing potency and enzyme kinetics. R.S., X.D., and F.W. conducted the animal experiments and qRT-PCR. L.L. and J.Z. evaluated the safety assessment data and H.F. supervised the study. Y.W., Y.F., Y.Z., R.Z., H.S., and J.X. collected the diffraction data and determined the complex structure. R.S., C.W., and J.Y. analyzed the data and wrote the manuscript.

## Competing interests
Y.D., R.S., C.W., and J.Y. are listed as inventors on pending patent applications for h11B11. The other authors declare that they have no competing interests.
