## [Peer Review File · Nature Communications]

Reviewers' Comments:

Reviewer #1:

Remarks to the Author:

The manuscript by Du et al describes the use of anti-ACE2 antibody to block the binding of SARS CoV-2 and provide effective therapeutic effect in vitro and in vivo.

I have several major comments to this manuscript:

1. The use of anti-ACE2 antibodies that blocks the entry of SARS CoV-2 was recently demonstrated by Hoffman et al (Cell, 2020) using polyclonal antibodies, as a proof-of-concept to this approach. Yet, this work was not cited or mentioned here, and thus the basis for the authors novel hypothesis (line 92) is somewhat problematic. Indeed, the current work use mAbs and demonstrate their activity in vivo, but the manuscript should be revised to include the previous work and to highlight the novel findings.

2. Another work that was not mentioned in this manuscript have tested the activity of small molecule (FDA approved) that binds to ACE2 and inhibit the binding of SARS CoV-2 (Wang et al, Cell Research, 13; 2021). I believe that by using FDA approved drug should be much more effective and safe for clinical use than using mAb. As above, the authors at least cite and discuss this previous work and elaborate about the unique contribution that this antibody will have for SARS CoV-2 therapy over the approved drug.

3. The binding of the Ab to ACE2 results in 50% inhibition of the enzyme activity. While the authors regard this inhibition as "...does not substantially affect..." (line 159) and "slightly affected" (line 304), I believe that it is highly significant inhibition. The activity of ACE2 in the body is highly controlled in order to maintain accurate homeostasis. To my opinion, inhibition of 50% of the enzymatic activity will have a dramatically effect on the physiology and might be toxic. The murine model that was tested in this work in inappropriate to study the toxicological effect of this antibody (I guess that they are using the K18-hACE2). The authors should test their antibody in mice model where the hACE2 actually replace the murine one or isolate a parallel anti-mACE2 antibody that also inhibit the enzymatic activity by 50% and test how it is affecting the mice physiological parameters.

I have several additional specific comments:

4. Line 73 – ADE was not seen for SARS CoV-2

5. Line 94 – broad-spectrum? Of what? The enzyme? The virus?

6. Line 128 – I believe that the actual affinity between RBD and ACE2 is much lower. We have measured affinity of 0.1nM (unpublished data). Please review the literature again for other studies, as I believe that different values that these mentioned in the manuscript were reported several times by many groups.

7. Line 133 – what is the basis for this assumption (the stability)?

8. Line 165 – what is the differences between these 39 VSVs?

9. Lines 193-194 – this number is not supported by the provided graph.

10. Line 199 – there is no data regarding the mice model that was used here (K18-hACE2?).

Please add

11. Lines 206-207 – Taking into account that K18-hACE2 model is used, then this infection model is not transient in nature. The study should include long-term evaluation (up to 14-21 days).

12. Line 212 – A 10-fold reduction in the viral load is not enough to protect the animals from death. The authors should repeat the experiment and test the viral load, animal weight and survival for a longer period of time in order to convince that this effect is indeed significant.

13. Line 234 – please provide the detailed graphical structure of the complex.

14. Line 240 – this data is not provided in fig. 4a or 4b.

15. Line2 243-245 – this point is not clear. There are multiple contact points between the antibody and the hACE2, thus it is not clear how the ablation of one of these will have such an effect on the binding.

16. Lines 261-263 – this sentence is not clear.

17. Lines 272-275 - this sentence is not clear.

18. Line 307 – "...promoted carboxypeptidase activity..." – it should be the other way around

19. Fig 1 and 2 – please indicate the concentration of what in the x-axis

20. Fig 2a, b – it seems that the linear regression fit is not set as Bmax=100%. Please change the fit parameters.

21. Fig. 4 – this figure is not informative. It should be re-organized.

Reviewer #2:

Remarks to the Author:

The manuscript by Du et al. reports the study of the activity of a humanized ACE2-blocking antibody h11B11 that can prevent SARS-CoV and SARS-CoV-2, but not hCoV-NL63, from binding to ACE2 thereby preventing and alleviating SARS-CoV-2 infection in a humanized mouse model. The search for various therapeutic options to treat COVID-19 is an important endeavor. The paper can be accepted after some minor revisions according to the comments summarized below.

The Results section is written as one large section, although it reports various disparate experiments. Thus, it might be better to create sub-heading in order to bring in some structure into the text.

The values of dissociation constants, K_D , should be given taking into consideration the correct number of significant figures. For example, 2.95 ± 1.02 should be given as 3 ± 1 , etc.

The last sentence of the second to last paragraph in Results makes no sense: "Meanwhile, the S19P variant eliminated the interaction between hACE2 and Mab h11B11 is structurally supported". First, if structurally supported, where is the structure, what structure is meant here? Second, the sentence is not comprehensible.

The usage of the organic chemistry term "stereospecific" is ill-defined when the authors describe binding of h11B11 to essentially the same region on hACE2 where SARS-CoV and SARS-CoV-2 RDB would bind. Do the authors follow convention of antibody research?

The authors need to comment on the quality of the X-ray data and on the quality of the structure that was refined at a very low resolution of 3.8Å. The overall R merge is very high at over 40%; is that normal for low-resolution X-ray structures. At least R_{pim} is reasonable.

Lastly, English needs to be improved throughout the manuscript and especially in Discussion.

Reviewer #3:

Remarks to the Author:

Du et al. describes the selection of an anti-hACE2 antibody that potently inhibits SARS-CoV-2 infection. This is a straightforward study, and the authors have done logical experiments that are necessary to demonstrate the utility of the antibody. These include the demonstration that the antibody neutralizes VSV pseudoviruses carrying major RBD mutations, that its binding to ACE2 is not influenced by any ACE2 polymorphism except one, and that it significantly reduces viral loads, interstitial pneumonia, leukocyte infiltration, and focal hemorrhage in the lung of infected mice. The reviewer has one scientific and one general concern.

Major:

1. Although the authors showed that hACE2 catalytic activity was not reduced by the binding of this antibody, the assay was conducted using soluble form of hACE2. Because the antibody is a dimer, it likely down-regulates cell-surface expressed hACE2, which will perhaps result in dysregulation of renin-angiotensin system, a similar condition caused by SARS-CoV-2 infection. Therefore, it is important to show hACE2 cell-surface expression level after the antibody is added and incubated at 37°C in a time course experiment. On the other hand, if a monomeric form of the antibody still efficiently inhibits pseudovirus or SARS-CoV-2 infection, ACE2 down-regulation is an irrelevant issue.
2. The authors can benefit from a deep editing by a native English speaker. There are many places where the meaning was unclear.

Minor:

1. Fig 1a, quantification with statistical significance should be shown
2. Because there are more than one type of hACE2-transgenic mouse lines, the authors need to describe the nature of the hACE2-transgenic mice used in this study, and especially identify the

promoter used.

3. Line 189: "transduction" is not proper term for live virus. It should be "infection".

We thank the three reviewers for their careful review of this manuscript. Each reviewer provided a number of constructive and thoughtful comments that were extremely helpful in revising this manuscript. We have now modified the manuscript according to their comments. Each point has been addressed.

The reviewers' comments are reproduced in their entirety in italics.

Reviewer #1 (Remarks to the Author):

1. The use of anti-ACE2 antibodies that blocks the entry of SARS CoV-2 was recently demonstrated by Hoffman et al (Cell, 2020) using polyclonal antibodies, as a proof-of-concept to this approach. Yet, this work was not cited or mentioned here, and thus the basis for the authors novel hypothesis (lines 92) is somewhat problematic. Indeed, the current work use mAbs and demonstrate their activity in vivo, but the manuscript should be revised to include the previous work and to highlight the novel findings.

Response: We thank the reviewer's suggestions. The reviewer is quite right that our description in our initial manuscript missed the important article. We have revised the manuscript in the introduction as follows (line 87-92):

“.....Ibalizumab, an approved MAb that binds human CD4 to block HIV-1 infection, showed antiviral and immunologic activity in a phase 3 study. Meanwhile, the results from Hoffman et al showed that mouse-derived polyclonal antibodies targeting ACE2 blocked SARS-CoV-2 pseudovirus-infected host cells. However, monoclonal antibody drugs targeting ACE2 have not been reported.....”

2. Another work that was not mentioned in this manuscript have tested the activity of small molecule (FDA approved) that binds to ACE2 and inhibit the binding of SARS CoV-2 (Wang et al, Cell Research, 13; 2021). I believe that by using FDA approved drug should be much more effective and safe for clinical use than using mAb. As above, the authors at least cite and discuss this previous work and elaborate about the

unique contribution that this antibody will have for SARS CoV-2 therapy over the approved drug.

Response: We thank the reviewer's comments and suggestions. We read and compared the results reported by Wang and our manuscript. In the preventive group, the viral load changes of dalbavancin-treating mice ($P < 0.01$ on 1 dpi and $p < 0.05$ on 3 dpi) (figure in the left panel) were significantly weaker than those of our Ab ($P < 0.0001$ in 5 mg/kg and 25 mg/kg) (figure in the right panel). The therapeutic effect was not evaluated in the mice model in the reference, so we cannot compare the therapeutic effects of dalbavancin and Ab.

We cited and discussed the studies mentioned by the reviewer in the revised manuscript (lines 326-332).

“.....During the revision of this article, dalbavancin, which has been approved for the treatment of acute bacterial skin and skin structure infections (ABSSSI) caused by designated susceptible gram-positive bacteria in adults, was reported to prevent SARS-CoV-2 infection in animal models. In the preventive group, the viral load changes of dalbavancin-treating mice were significantly weaker than those of MAb h11B11 (Fig. 3a). Combined with our results, these reports suggest that antibodies are more potent than dalbavancin as countermeasures against SARS-CoV-2.....”

3. The binding of the Ab to ACE2 results in 50% inhibition of the enzyme activity.

While the authors regard this inhibition as "...does not substantially affect..." (line 159) and "slightly affected"(line 304), I believe that it is highly significant inhibition. The activity of ACE2 in the body is highly controlled in order to maintain accurate homeostasis. To my opinion, inhibition of 50% of the enzymatic activity will have a dramatic effect on the physiology and might be toxic. The murine model that was tested in this work is inappropriate to study the toxicological effect of this antibody (I guess that they are using the K18-hACE2). The authors should test their antibody in a mice model where the hACE2 actually replaces the murine one or isolate a parallel anti-mACE2 antibody that also inhibits the enzymatic activity by 50% and test how it is affecting the mice physiological parameters.

Response: We are thankful for the reviewer's comments. We are sorry that we missed some important description in the setting of the control group. To eliminate the non-specific effect of protein on enzyme activity, we set up an isotype IgG group as negative control except PBS group. We found that compared to isotype control Ab, mAb h11B11 didn't inhibit the catalytic activity of ACE2 in vitro (Fig. 1d). Furthermore, the enzymatic kinetic constants of hACE2 in the presence of h11B11 and isotype IgG are almost equal (Supplementary Table 1).

In this study, we used the homozygous hACE2 mice with murine ACE2 receptors which were inactivated (Sun et al, *Cell Host and Microbe*, 2020 Jul 8;28(1):124-133.e4). We add the description of the hACE2 mice genotype in the revised manuscript (lines 212-214).

".....Homozygous mice expressing hACE2 were generated by using CRISPR/Cas9 to knock the full cDNAs of hACE2 into exon 2, the first coding exon, of the mAce2 gene located at GRCm38.p6....."

During the revision of this manuscript, we determined the toxicity and the Maximum Tolerated Dose (MTD) of this Ab to cynomolgus monkeys. 4 animals were administered via repeated intravenous infusion with doses of 60 and 180 mg/kg. All animals survived until the planned necropsy. Compared with the pre-dose, 4 animals

showed stable blood pressure even after the second dosage. There were no test article-related abnormalities in clinical observation, bodyweight, and lymphocyte. Therefore, we confirm the MTD of the Ab was 180 mg/kg in cynomolgus monkeys and supplement relevant data in the article (lines 259-286 and Fig. 4).

“.....To assay the safety of h11B11 *in vivo*, cynomolgus monkeys were employed to explore the toxicity and the maximum tolerated dose (MTD). An SPR assay confirmed that this MAb shows analogous binding affinities to hACE2 and cynomolgus ACE2 (Supplementary Fig. 4). Four male cynomolgus monkeys (animal No. 1001, 1002, 1101, and 1102) were randomly divided into two groups according to body weight. Cynomolgus monkeys in the high-dose group (180 mg/kg to 1101 and 1102) or the low-dose group (60 mg/kg to 1001 and 1002) were administered via repeated intravenous infusion (once a week for three weeks) at a dosage of 1 ml/kg at a rate of 1 ml/min. Evaluation indicators included clinical observation, blood pressure, and clinical pathology (haematology and serum biochemistry). During the study, the

animals in each group survived until the planned necropsy. There were no test article-related abnormalities in clinical observations, body weight, or food consumption in the 60 or 180 mg/kg groups. To enable animals to adapt to the vest used for measuring blood pressure and reflect the situation after multiple administrations, we monitored the changes in blood pressure after the second administration. Compared with the predose measurements, the mean diastolic and systolic blood pressure of all animals in the following six days after the second dosing showed only minor changes, from 90 to 102 mm Hg (Fig. 4e and f). The fluctuation of blood pressure in the low-dose group was smaller (from 96 to 101 mm Hg) (Fig. 4e) than that of high-dose injected animals (from 90 to 102 mm Hg) (Fig. 4f). Meanwhile, the haematology assays showed that the white blood cell (WBC), red blood cell (RBC), monocyte (MONO), and lymphocyte (LYMPH) counts of all cynomolgus monkeys were equally stable at 21 days (Fig. 4g). To evaluate the toxicity of h11B11 to animals, aspartate transaminase (AST), alanine transaminase (ALT), C-reactive protein, and total bilirubin (TBIL) were assayed. These indexes of the animals in the two groups were not significantly affected by multiple administrations (Fig. 4h). Therefore, multiple dosages of h11B11 were observed to be safe in this preclinical study, and the MTD was 180 mg/kg.....”

I have several additional specific comments:

4. Line 73 – ADE was not seen for SARS CoV-2

Response: We are thankful for the reviewer's comments and agree with the reviewer on the viewpoint. No obvious ADE effect is found in the clinical trials. We deleted this description in the revised manuscript.

5. Line 94 – broad-spectrum? Of what? The enzyme? The virus?

Response: We are sorry that the description is not clear enough and might have caused some misunderstandings. We mean that MAbs blocking hACE2 could effectively neutralize SARS-CoV, SARS-CoV-2, and the mutants of SARS-CoV-2 (Fig. 2). We revised the sentence in lines 92-95.

“.....In this study, we identify a broad-spectrum humanized ACE2/RBD-blocking MAb, h11B11, which demonstrates potent inhibitory activity against SARS-CoV and circulating global SARS-CoV-2 lineages *in vitro*.....”

6. Line 128 – I believe that the actual affinity between RBD and ACE2 is much lower. We have measured affinity of 0.1nM (unpublished data). Please review the literature again for other studies, as I believe that different values that these mentioned in the manuscript were reported several times by many groups.

Response: We are thankful for the reviewer's comments. Generally, there are some differences in the affinity (K_D) of SARS-CoV-2-RBD and ACE2 from different labs. We found that the K_D varies from 36.4 to 133.3 for RBDS/hACE2 and ~15 nM for S/hACE2 binding (Wrapp, D. et al. Science 367, 1260-1263, doi:10.1126/science.abb2507, Yurkovetskiy, L. et al. Cell 183, 739-751.e738, doi:10.1016/j.cell.2020.09.032 and Yan, R. et al. Science 367, 1444-1448, doi:10.1126/science.abb2762). In our lab, we measured the affinity of 133.3 nM (Wang, Q. et al. Cell 181, 894-904.e899. doi:10.1016/j.cell.2020.03.045). We revised the sentence in the revised manuscript (in lines 127-130).

“.....SPR showed that h11B11 has a higher affinity ($K_D=2.95$ nM) than the SARS-CoV-2-RBD or SARS-CoV-2 S protein for immobilized hACE2 ($K_D=36.4-133.3$ nM

for SARS-CoV-2-RBD and ~15 nM for SARS-CoV-2 S), almost entirely due to a slower off-rate.....”

7. Line 133 – what is the basis for this assumption (the stability)?

Response: N-terminal helix (NTH) of ACE2 is the binding site of SARS-CoV-2-RBD. Since h11B11 can block the interaction between ACE2 and RBD, we conjecture h11B11 also binds to NTH of ACE2 and the variations in NTH of ACE2 may affect the binding stability of h11B11 Ab and ACE2 variants.

8. Line 165 – what is the differences between these 39 VSVs?

Response: We are thankful for the reviewer’s comments. All the pseudoviruses were produced using the VSV delta G/luciferase system with original or mutant S proteins. These S coding sequences include the original SARS-CoV-2, SARS-CoV-2 sustained transmission lineages, and high-frequency mutants from GISAID, and we added the description in lines 174-176.

“.....These S coding sequences include the original SARS-CoV-2, SARS-CoV-2 sustained transmission lineages, and high-frequency mutants from GISAID.....”

9. Lines 193-194 – this number is not supported by the provided graph.

Response: We agree with reviewer’s comments. We detected the neutralizing activity of the h11B11 against HCoV-NL63 at the concentration of 100 µg/ml and didn’t obtain the positive results. We revised the manuscript and showed the graph in supplementary Fig. 3.

10. Line 199 – there is no data regarding the mice model that was used here (K18-hACE2?). Please add

Response: We are sorry for missing the detailed descriptions of the mice model. In this study, we didn't use the K18-hACE2 mice for the in vivo experiments, and we used the hACE2 mice which were described previously (Sun et al, *Cell Host and Microbe*, 2020 Jul 8;28(1):124-133.e4). We add the description of hACE2 mice in the revised manuscript (lines 212-214).

“.....Homozygous mice expressing hACE2 were generated by using CRISPR/Cas9 to knock the full cDNAs of hACE2 into exon 2, the first coding exon, of the mAce2 gene located at GRCm38.p6.....”

11. Lines 206-207 – Taking into account that K18-hACE2 model is used, then this infection model is not transient in nature. The study should include long-term evaluation (up to 14-21 days).

Response: We are thankful for the reviewer's suggestion. We didn't use K18-hACE2 mice in this study, and we used hACE2 mice which were described in the *Cell Host and Microbe* paper (Sun et al, *Cell Host and Microbe*, 2020 Jul 8;28(1):124-133.e4.). As observed in this animal model, mice infected with SARS-CoV-2 displayed transient infection and viral replication, the peak of viral titres was found at 5-7 dpi (the below figure from the reference), and the viral load will naturally decline. Meanwhile, inoculation of SARS-CoV-2 does not lead to lethal effects in this mice model. So the study does not include a long-term evaluation.

12. Line 212 – A 10-fold reduction in the viral load is not enough to protect the animals from death. The authors should repeat the experiment and test the viral load, animal weight and survival for a longer period of time in order to convince that this effect is indeed significant.

Response: We are thankful for the reviewer’s comments and suggestions. We checked the literature and found that several of the studies did show a better therapeutic effect in terms of viral load after antibody or drug treatment (Yunlong Yao, et al. *Cell* 182, 1-12, July, 2020; Gan Wang, et al. *Cell Res.* 2021 Jan;31(1):17-24). While, we find that they treated mice with antibody 2 hours after viral infection (Yunlong Yao, et al. *Cell* 182, 1-12, July, 2020), or treated mice with the drug at the same time of viral infection (Gan Wang, et al. *Cell Res.* 2021 Jan;31(1):17-24). We treated mice 24 hours after infection for the treatment group, and in this condition, the virus already infected and amplified in the lung for 24 hours, so the basal virus load was already high before h11B11 treatment, and we believe that this could be the reason that the MAb h11B11 treatment only caused 10 folds reduction for the viral load in the treatment group.

In the preventive group, the titres of 9/10 animals were below detections (Fig. 3a), which had a comparable effect to the published studies. Even in the treating groups, there are significant differences between MAb-treating and placebo-treating mice ($P < 0.01$ in 5 mg/kg and $p < 0.001$ in 25 mg/kg). So, we believe the MAb h11B11 is an effective candidate molecule against COVID-19.

As reported previously (Sun et al, *Cell Host and Microbe*, 2020 Jul 8;28(1):124-

133.e4), this hACE2 mice strain infected with SARS-CoV-2 displayed transient infection, and inoculation of SARS-CoV-2 does not lead to lethal effect in these mice model.

Due to the characteristics of the mice model and the limited capacity of the biosafety level-3 facility, we could not repeat the experiment in a short time. In the future, we will repeat the experiment via an additional model.

We also checked some literature, and found that most of the observation time for hACE2 mice SARS-CoV-2 infection studies is within 7 dpi:

1. Yunlong Yao, et al. *Cell* 182, 1-12, July, 2020;
2. Gan Wang, et al. *Cell Res.* 2021 Jan;31(1):17-24;
3. Linlin Bao, et al. *Nature.* Vol 583, 30 July 2020, 831-833;
4. Bin Zhou, et al. *Nature.* Vol 592, 1 April 2021, 122-127.

13. Line 234 – please provide the detailed graphical structure of the complex.

Response: We are sorry for our insufficient description of the detailed graphical structure. Although we have tried to prepare more protein complexes in crystallization conditions, the structure of the h11B11/hACE2 complex was determined at a resolution of 3.8 Å. Some of contacts and interactions such as hydrogen bonds cannot be accurately marked due to the low resolution. Yet, a cutoff value of 4.5 Å is used to define the interaction sites of Ab and ACE2. We add the description in the revised manuscript (in lines 294-297) and the Fig. 5.

“.....For further analysis of possible interactions, a cutoff value of 4.5 Å is used to define the main contacts between h11B11 and hACE2.....”

14. Line 240 – this data is not provided in fig. 4a or 4b.

Response: We are thankful for the reviewer’s comments. Some of contacts and interactions such as hydrogen bonds cannot be accurately marked due to the low resolution. We revised the description in lines 294-297 and the figure.

“.....For further analysis of possible interactions, a cutoff value of 4.5 Å is used to

define the main contacts between h11B11 and hACE2. All three CDRs of the VH and LCDR1 and LCDR3 provide contacts with hACE2, while LCDR2 is not engaged (Fig. 5a).....”

15. Line2 243-245 – this point is not clear. There are multiple contact points between the antibody and the hACE2, thus it is not clear how the ablation of one of these will have such an effect on the binding.

Response: We thank for the reviewer’s comments and suggestion. Although multiple contacts co-influence the binding affinity, a single key residue mutation could ablate the binding between MAbs and antigens. For example, E484K on SARS-CoV-2-RBD completely or markedly abolished the activities of MAb 2-15 and C121(Wang, P. et al. Nature 593, 130-135, doi:10.1038/s41586-021-03398-2). In this study, we found that MAb h11B11 isn’t able to bind to hACE2 with S19P mutation. Actually, we aren’t sure the mechanism for that due to the low resolution of structure. We speculate two possibilities to affect the binding and add the description in the revised manuscript in lines 360-367.

“.....We speculate two reasons for the S19P variant eliminating the interaction between hACE2 and MAb h11B11. Although multiple contacts co-influencing the binding affinity, a single key site mutation could ablate the binding between MABs and antigens. For example, E484K on SARS-CoV-2-RBD completely or markedly abolished the activities of MAb 2-15 and C121. Another possibility is that the proline substitution could destroy the conformation of N-terminal helix, which leads to the disappearance of interaction.....”

16. Lines 261-263 – this sentence is not clear.

Response: We are thankful for the reviewers' comments. We revised the sentence as “.....No structural evidence directly indicates that the MAb h11B11 inhibits the binding of HCoV-NL63-RBD and ACE2” (lines 314-315).....”

17. Lines 272-275 - this sentence is not clear.

Response: We are thankful for the reviewers' comments. We deleted the ambiguous sentence “.....Most importantly, the ACE2 receptor-specific distribution and enrichment of h11B11 MAb may also have improved bioavailability compared with that of a relatively uniform distribution of ACE2-Fc fusion protein in the prophylactic setting.....”

18. Line 307 – “..promoted carboxypeptidase activity...”– it should be the other way around

Response: We thank for the reviewer's comments. As mentioned above, we set up two negative control groups. We compared the activities between h11B11 and isotype IgG group in the revised manuscript. Incubated with h11B11, the carboxypeptidase activity of hACE2 was promoted compared to hACE2 alone in vitro and maintained comparable to isotype IgG (Fig. 1d and Supplementary Table 1).

19. Fig 1 and 2 – please indicate the concentration of what in the x-axis

Response: We are thankful for the reviewer's comment and the value in the x-axis of Fig. 1 and 2 represents the concentrations of substrate (Fig. 1d) and the concentrations of antibody (Fig. 2a, b, and d), and we clarified them in the figure legends of the revised manuscript.

20. Fig 2a, b – it seems that the linear regression fit is not set as Bmax=100%. Please change the fit parameters.

Response: We are thankful for the reviewers' suggestions. As we presented the data as mean \pm SEM in Fig. 2a-b and some of the data/arrow bar will be out of axis limitation if Bmax is set as 100%.

21. Fig. 4 – this figure is not informative. It should be re-organized.

Response: We thank for the reviewer's comments and suggestions. We revised the the figure 5 (supplemented in reviewer' comment 14) to provide more informations.

Reviewer #2 (Remarks to the Author):

The manuscript by Du et al. reports the study of the activity of a humanized ACE2-blocking antibody h11B11 that can prevent SARS-CoV and SARS-CoV-2, but not hCoV-NL63, from binding to ACE2 thereby preventing and alleviating SARS-CoV-2 infection in a humanized mouse model. The search for various therapeutic options to treat COVID-19 is an important endeavor. The paper can be accepted after some minor revisions according to the comments summarized below.

The Results section is written as one large section, although it reports various disparate experiments. Thus, it might be better to create a sub-heading to bring in some structure into the text.

Response: We are thankful for the reviewer's suggestion, and we created sub-headings in the revised manuscript.

The values of dissociation constants, K_D , should be given taking into consideration the correct number of significant figures. For example, 2.95 ± 1.02 should be given as 3 ± 1 , etc.

Response: We thank the reviewer's suggestion. We checked the literature and followed the presenting format adopted by some literature and the number of digits of K_D in this manuscript is consistent.

1. Sisi Kang, *et al.* A SARS-CoV-2 antibody curbs viral nucleocapsid protein-induced complement hyperactivation. *Nat Commun.* 2021 May 11;12(1):2697.
2. Shi R, *et al.* A human neutralizing antibody targets the receptor-binding site of SARS-CoV-2. *Nature.* 2020 Aug;584(7819):120-124.
3. Chi, X. *et al.* A neutralizing human antibody binds to the N-terminal domain of the Spike protein of SARS-CoV-2. *Science* 369, 650-655, doi:10.1126/science.abc6952 (2020).

The last sentence of the second to the last paragraph in Results makes no sense: "Meanwhile, the S19P variant eliminated the interaction between hACE2 and Mab h11B11 is structurally supported". First, if structurally supported, where is the structure, what structure is meant here? Second, the sentence is not comprehensible.

Response: Reviewer 2 raised similar questions as reviewer1 regarding to our careless insufficient description. The reviewer is right about the mechanism of S19P ablated the binding is an overstatement saying. We speculate two mechanisms to affect the binding and add the description in the revised manuscript in lines 360-367.

".....We speculate two reasons for the S19P variant eliminating the interaction between hACE2 and MAb h11B11. Although multiple contacts co-influencing the binding affinity, a single key site mutation could ablate the binding between MAbs and antigens. For example, E484K on SARS-CoV-2-RBD completely or markedly abolished the activities of MAb 2-15 and C121. Another possibility is that the proline substitution could destroy the conformation of N-terminal helix, which leads to the disappearance of interaction....."

We depleted this sentence “Meanwhile, the S19P variant eliminated the interaction between hACE2 and MAb h11B11 is structurally supported” in the revised manuscript.

The usage of the organic chemistry term “stereospecific” is ill-defined when the authors describe binding of h11B11 to essentially the same region on hACE2 where SARS-CoV and SARS-CoV-2 RDB would bind. Do the authors follow convention of antibody research?

Response: We are thankful for the reviewers’ suggestion, and we deleted “stereospecific” in the revised manuscript. We followed the convention of antibody research.

The authors need to comment on the quality of the X-ray data and on the quality of the structure that was refined at a very low resolution of 3.8Å. The overall R merge is very high at over 40%; is that normal for low-resolution X-ray structures. At least Rpim is reasonable.

Response: We are thankful for the reviewer’s comment. R_{merge} is an unweighted statistic that describes how well the unmerged equivalent observations agree with one another and is related to a number of factors including high data redundancy, instrumental factor, X-ray decay, etc. It is arguably not the best indicator for data quality, compared to for example I/σ and CC1/2. There are many discussions on this topic, for example at the 2012 CCP4 Study Weekend (<https://journals.iucr.org/d/issues/2013/07/00/>). Since we do have high data redundancy (13.0 overall, 13.4 last shell), we decided to consider all the data quality indicators to cut off our data at 3.8 angstroms, instead of relying only on R_{merge}. In particular, we have very good CC1/2 at last shell (a portion of the HKL2000 scale log file is shown below), indicating high data quality. Also, we obtained very reasonable R_{work}/R_{free} (0.276 / 0.327) after Rosetta refinement in Phenix. Therefore, we believe our data cutoff is reasonable.

Shell limit	Lower Angstrom	Upper Angstrom	Average I	Average error	Average stat.	Norm. Chi**2	Linear R-fac	Square R-fac	Rmeas	Rpim	CC1/2	CC*
50.00	10.29	10.29	50.9	2.0	1.3	0.964	0.092	0.098	0.097	0.029	0.997	0.999
10.29	8.18	8.18	39.9	1.9	1.4	0.516	0.093	0.086	0.097	0.027	0.998	0.999
8.18	7.15	7.15	17.9	1.6	1.4	0.466	0.183	0.156	0.190	0.052	0.995	0.999
7.15	6.49	6.49	12.1	1.7	1.6	0.462	0.291	0.209	0.303	0.085	0.993	0.998
6.49	6.03	6.03	8.1	1.7	1.7	0.458	0.451	0.353	0.469	0.128	0.977	0.994
6.03	5.67	5.67	7.8	1.8	1.8	0.452	0.486	0.370	0.505	0.138	0.973	0.993
5.67	5.39	5.39	9.0	2.1	2.0	0.461	0.445	0.330	0.465	0.133	0.978	0.995
5.39	5.16	5.16	9.5	2.1	2.1	0.489	0.490	0.354	0.511	0.142	0.978	0.994
5.16	4.96	4.96	9.3	2.1	2.1	0.511	0.514	0.356	0.534	0.146	0.977	0.994
4.96	4.79	4.79	9.9	2.2	2.2	0.502	0.510	0.297	0.530	0.144	0.989	0.997
4.79	4.64	4.64	9.0	2.2	2.2	0.507	0.561	0.393	0.583	0.157	0.973	0.993
4.64	4.50	4.50	8.8	2.3	2.3	0.493	0.590	0.337	0.613	0.165	0.983	0.996
4.50	4.39	4.39	8.4	2.4	2.4	0.493	0.646	0.383	0.671	0.180	0.980	0.995
4.39	4.28	4.28	7.5	2.4	2.4	0.502	0.729	0.503	0.757	0.203	0.958	0.989
4.28	4.18	4.18	5.8	2.5	2.5	0.464	0.930	0.606	0.967	0.261	0.941	0.985
4.18	4.09	4.09	5.7	2.6	2.6	0.452	0.948	0.591	0.989	0.278	0.950	0.987
4.09	4.01	4.01	5.1	2.6	2.6	0.462	1.054	0.644	1.098	0.305	0.942	0.985
4.01	3.94	3.94	4.7	2.6	2.6	0.449	1.200	0.718	1.249	0.342	0.927	0.981
3.94	3.87	3.87	5.5	2.8	2.8	1.016	1.268	1.342	1.324	0.378	0.858	0.961
3.87	3.80	3.80	3.5	2.6	2.6	0.544	1.739	0.987	1.808	0.493	0.909	0.976
All reflections			12.2	2.2	2.1	0.528	0.415	0.258	0.433	0.119		

Lastly, English needs to be improved throughout the manuscript and especially in Discussion.

Response: We are thankful for the reviewer's suggestion, and the manuscript was revised by a professional language editing service agency during revision.

Reviewer #3 (Remarks to the Author):

Du et al. describes the selection of an anti-hACE2 antibody that potently inhibits SARS-CoV-2 infection. This is a straightforward study, and the authors have done logical experiments that are necessary to demonstrate the utility of the antibody. These include the demonstration that the antibody neutralizes VSV pseudoviruses carrying major RBD mutations, that its binding to ACE2 is not influenced by any ACE2 polymorphism except one, and that it significantly reduces viral loads, interstitial pneumonia, leukocyte infiltration, and focal hemorrhage in the lung of infected mice. The reviewer has one scientific and one general concern.

Major:

1. Although the authors showed that hACE2 catalytic activity was not reduced by the binding of this antibody, the assay was conducted using soluble form of hACE2. Because the antibody is a dimer, it likely down-regulates cell-surface expressed

hACE2, which will perhaps result in disregulation of renin-angiotensin system, a similar condition caused by SARS-CoV-2 infection. Therefore, it is important to show hACE2 cell-surface expression level after the antibody is added and incubated at 37°C in a time course experiment. On the other hand, if a monomeric form of the antibody still efficiently inhibits pseudovirus or SARS-CoV-2 infection, ACE2 down-regulation is an irrelevant issue.

Response: We are thankful for the reviewer's comment. During the revision of this manuscript, we followed the reviewer's suggestions and examined whether h11B11 MAb would change the cell surface expressional level of ACE2. We didn't find h11B11 MAb had any impact on the cell surface expression of ACE2 by membrane-cytoplasm fractionation and flow cytometry experiments. We supplemented relevant data in the article (lines 246-257 and Fig. 4a-d).

“.....Antibodies could induce the internalization and degradation of the targeting protein on the cell surface, so we sought to examine whether incubation of MAb h11B11 had any impact on the expressional level of ACE2 on the cell surface. HEK293T-hACE2 cells were incubated with MAb h11B11 at 37°C to allow internalization and ACE2 protein level was examined by immunoblot and flow cytometry analysis. We found that the hACE2 protein level remained unchanged on the cell membrane after incubation with MAb h11B11 at a high concentration of 10 µg/ml (Fig. 4a, b). This result was further confirmed by flow cytometry analysis, that hACE2 level on the cell surface was consistent in 4 hours or 24 hours after incubation with a series of h11B11 protein (Fig. 4c, d). Therefore, the levels of membrane-expressed hACE2 were not affected after incubating with h11B11, which indicates the safety of this MAb.....”

We also determined the toxicity and the Maximum Tolerated Dose (MTD) of this

MAb to cynomolgus monkeys, confirm the MTD of the Ab was 180 mg/kg in cynomolgus monkeys, and supplement relevant data in the article (lines 258-285 and Fig. 4e-h).

2. The authors can benefit from a deep editing by a native English speaker. There are many places where the meaning was unclear.

Response: We are thankful for the reviewer's suggestion and the manuscript was revised by a professional language editing service agency during revision.

Minor:

1. Fig 1a, quantification with statistical significance should be shown

Response: The reviewer is right that the frequency of positive clusters within the cell must be compared. We revised Fig. 1a according to the reviewer's suggestions.

2. Because there is more than one type of hACE2-transgenic mouse lines, the authors need to describe the nature of the hACE2-transgenic mice used in this study, and especially identify the promoter used.

Response: We are thankful for the reviewer's suggestion. Reviewer3 raised similar questions as reviewer1 regarding our careless insufficient description of the mice model. In this study, we used the homozygous hACE2 mice with murine ACE2 receptors that were inactivated (Sun et al, *Cell Host and Microbe*, 2020 Jul 8;28(1):124-133.e4). We add the description of the hACE2 mice genotype in the revised manuscript (lines 212-214).

“.....Homozygous mice expressing hACE2 were generated by using CRISPR/Cas9 to knock the full cDNAs of hACE2 into exon 2, the first coding exon, of the mAce2 gene located at GRCm38.p6.....”

3. Line 189: “transduction” is not proper term for live virus. It should be “infection”.

Response: We are thankful for the reviewers' suggestion and sorry about the inaccurate description. We revised the description according to the reviewer's

suggestion (line 197).

Reviewers' Comments:

Reviewer #1:

Remarks to the Author:

The authors have addressed all my comments

Reviewer #2:

Remarks to the Author:

The authors carefully considered the comments of this reviewer and addressed them sufficiently well.

Reviewer #3:

Remarks to the Author:

The authors have conducted the experiments this reviewer suggested, although the result is rather surprising. An explanation for why or how h11B11, a dimeric antibody, did not down-regulate ACE2 would have helped. Although English editing improved the readability of the manuscript, there still are quite a few places where meaning is unclear.

Reviewer #4:

Remarks to the Author:

The manuscript entitled "A humanized ACE2-targeting antibody blocks the entry of SARS-CoV, SARS-CoV-2 and variants" by Du et al., characterizes an ACE2-blocking monoclonal antibody. I was asked to review the NHP safety data and this review will be limited to these areas. Overall the approach to the safety testing was well designed and described in the manuscript. I do have a few comments that may strengthen the paper.

Major.

Lines 262-264. Why were only male cynos used? Both sexes are usually represented in safety testing studies. While this probably is not an issue in this scenario, but the use of males only should be discussed.

Lines 264-267. How were the doses chosen? Is this based on what a human would receive?

Line 269. The necropsy findings should be presented. If there was no findings, please state. Was any histology done? If yes, please include. If no, please discuss. Histological evaluation of tissues will pick up on safety issues missed during necropsy and is usually included in safety evaluation studies.

Minor.

Line 246 change "could" to "can"

We thank the reviewer for the careful review of this manuscript. The reviewer provided a number of constructive and thoughtful comments that were extremely helpful in revising this manuscript. We have now modified the manuscript according to the comments. Each point has been addressed.

The reviewer's comments are reproduced in their entirety in italics.

Reviewer #4 (Remarks to the Author):

The manuscript entitled "A humanized ACE2-targeting antibody blocks the entry of SARS-CoV, SARS-CoV-2 and variants" by Du et al., characterizes an ACE2-blocking monoclonal antibody. I was asked to review the NHP safety data and this review will be limited to these areas. Overall the approach to the safety testing was well designed and described in the manuscript. I do have a few comments that may strengthen the paper.

Major.

Lines 262-264. Why were only male cynos used? Both sexes are usually represented in safety testing studies. While this probably is not an issue in this scenario, but the use of males only should be discussed.

Response: We are thankful for the reviewer's comments. Yes, only male cynos were used. Considering biologics generally showed no sexual difference in drug exposure and safety, and due to the increasingly limited animal resources, we use single sex in this preliminary toxicity study. And in the future GLP-compliant toxicity study, we will use both sexes to comprehensively evaluations. We have revised the manuscript in the introduction as follows (line 375-377):

".....The biologics generally showed no sexual difference in drug exposure and safety. Consequently, we employed single-sex animals in this preliminary toxicity study....."

Lines 264-267. How were the doses chosen? Is this based on what a human would receive?

Response: We are thankful for the reviewer's comment. For dose selection, in vivo efficiency of h11B11 in protecting ACE2 humanized mice from SARS-CoV-2 infection, 5 mg/kg or 25 mg/kg mAb h11B11 is efficacious in both prophylactic and treatment models. Clinical trials of antibody drugs related to COVID-19 have adopted a single administration research scheme, and the mAb dosages have been less than 50 mg/kg. Meanwhile, The drug concentration is 18 mg/mL, and maximal dose volume for repeated toxicity is 10 mL/kg, then the maximal dose is set as 180 mg/kg, which is 7.2 folds for in vivo study efficiency dose and 3.6 folds for clinical trial dose.

Line 269. The necropsy findings should be presented. If there was no findings, please state. Was any histology done? If yes, please include. If no, please discuss. Histological evaluation of tissues will pick up on safety issues missed during necropsy and is usually included in safety evaluation studies.

Response: We thank the reviewer's comments and suggestions. After scheduled euthanasia, gross pathology has been done, #1001 animal in 60 mg/kg and #1101 animal in 180 mg/kg had multifocal spleen raised with white discoloration on the cut surface, and #1101 animal had reduced thymus. Histology pathology had not been conducted. Considering the limited animal numbers and no dose response, the relevance of the abnormal findings of gross pathology and test-article related was not clearly. And in the future GLP-compliant toxicity study, we will evaluate drug tox comprehensively. We discussed the studies mentioned by the reviewer in the revised manuscript (lines 377-383).

“.....After scheduled euthanasia, gross pathology was performed. No. 1001 animal in the 60 mg/kg group and No. 1101 animal in the 180 mg/kg group had multifocal spleen raised with white discoloration on the cut surface. Meanwhile, No. 1101 animal showed reduced thymus. Considering the limited animal numbers and no dose-dependent

toxicity, the relevance of the abnormal findings of gross pathology and test-article related was not clear.....”

Minor.

Line 246 change “could” to “can”

Response: We are sorry for our inappropriate description. We revised the word in the manuscript.